# Mitochondrial ETF insufficiency drives neoplastic growth by selectively optimizing cancer bioenergetics

David Papadopoli[1,2]*, Ranveer Palia[1,3], Predrag Jovanovic[1,3], Sébastien Tabariès[4], Emma Ciccolini[1,3], Valerie Sabourin[1], Sebastian Igelmann[5], Shannon McLaughlan[6], Lesley Zhan[6], HaEun Kim[3], Nabila Chekkal[6], Krzysztof J Szkop[7], Thierry Bertomeu[8], Jibin Zeng[1], Julia Vassalakis[9], Farzaneh Afzali[9], Slim Mzoughi[10], Ernesto Guccione[10], Mike Tyers[11,12], Daina Avizonis[4,13], Ola Larsson[7], Lynne-Marie Postovit[9], Sergej Djuranovic[14,15], Josie Ursini-Siegel[1,2,3,6], Peter M Siegel[3,4,6], Michael Pollak[1,2,3]*, Ivan Topisirovic[2,3,6]*

[1]Lady Davis Institute, SMBD JGH, McGill University, Montréal, Canada; [2]Gerald Bronfman Department of Oncology, McGill University, Montréal, Canada; [3]Department of Experimental Medicine, McGill University, Montréal, Canada; [4]Rosalind and Morris Goodman Cancer Institute, McGill University, Montréal, Canada; [5]VIB Center for Cancer Biology, Department of Oncology, KU Leuven and Leuven Cancer Institute (LKI), Leuven, Belgium; [6]Department of Biochemistry, McGill University, Montréal, Canada; [7]Department of Oncology-Pathology, Science for Life Laboratory, Karolinska Institute, Stockholm, Sweden; [8]Institute for Research in Immunology and Cancer, Université de Montréal, Montréal, Canada; [9]Department of Biomedical and Molecular Sciences, Queen's University, Kingston, Canada; [10]Center of OncoGenomics and Innovative Therapeutics (COGIT), Department of Oncological Sciences, Tisch Cancer Institute, Icahn School of Medicine at Mount Sinai, New York, United States; [11]Program in Molecular Medicine, The Hospital for Sick Children, Toronto, Canada; [12]Department of Molecular Genetics, University of Toronto, Toronto, Canada; [13]Metabolomics Innovation Resource, McGill University, Montréal, Canada; [14]Department of Cell Biology and Physiology, Washington University School of Medicine, St. Louis, United States; [15]Department of Molecular Biology, Cell Biology and Biochemistry, Brown University, Providence, United States

*For correspondence:
david.papadopoli@mail.mcgill.ca (DP);
michael.pollak@mcgill.ca (MP);
ivan.topisirovic@mcgill.ca (IT)

## eLife Assessment

The authors present an **important** set of data implicating ETFDH as an epigenetically suppressed gene in cancer with tumor suppressive functions. The evidence is **convincing**, with the authors demonstrating that suppression of ETFDH activity results in accumulation of amino acids that impact metabolism via hyperactive mTORC1.

**Abstract** Mitochondrial electron transport flavoprotein (ETF) insufficiency causes metabolic diseases known as a multiple acyl-CoA dehydrogenase deficiency (MADD). In contrast to muscle, ETFDH is a non-essential gene in acute lymphoblastic leukemia NALM6 cells, and its expression is reduced across human cancers. In various human cancer cell lines and mouse models, ETF insufficiency caused by decreased ETFDH expression limits flexibility of OXPHOS fuel utilisation but paradoxically increases bioenergetics and accelerates neoplastic growth via activation of the mTORC1/

BCL-6/4E-BP1 axis. Collectively, these findings reveal that while ETF insufficiency is rare and has detrimental effects in non-malignant tissues, it is common in neoplasia, where ETFDH downregulation leads to bioenergetic and signaling reprogramming that accelerates neoplastic growth.

## Introduction

The mitochondrial electron transport chain (ETC) transfers electrons from reducing equivalents (NADH and $FADH_2$) towards complexes I/II, or from peripheral metabolic enzymes that reduce coenzyme Q (ubiquinone, Q) to ubiquinol ($QH_2$) (*Banerjee et al., 2022*). The electron transfer flavoproteins (ETFA and ETFB) accept electrons generated by the catabolism of branched-chain amino acids and fatty acids (*Mereis et al., 2021*; *Ghisla and Thorpe, 2004*; *Zhang et al., 2019*). ETFs interact with ETF dehydrogenase (ETFDH or ETF-ubiquinone oxidoreductase) (*El-Gharbawy and Vockley, 2018*), which controls electron flow towards complex III (*Figure 1—figure supplement 1A*). Loss-of-function mutations in ETF/ETFDH occur in metabolic diseases known as multiple acyl-CoA dehydrogenase deficiency (MADD) (*Olsen et al., 2007*; *Missaglia et al., 2021*). ETFDH is also essential for complex III activity in skeletal muscle (*Herrero Martín et al., 2024*). Surprisingly, we observed that in contrast to muscle, ETFDH is one of the most non-essential metabolic genes in cancer cells. Paradoxically, ETFDH downregulation leads to enhanced mitochondrial biogenesis and increased bioenergetic capacity of cancer cells. ETF insufficiency resulting from reduction in ETFDH levels triggers a trade-off favored by cancer cells, whereby reduced flexibility in fuel utilization for oxidative phosphorylation is offset by signaling from mitochondria to mTORC1 that remodels cancer cell bioenergetics to augment neoplastic growth.

## Results

### *ETFDH* abrogation accelerates tumor growth

ETFDH is essential in skeletal muscle cells (*Herrero Martín et al., 2024*). Intriguingly, a CRISPR-Cas9 knockout (KO) screen performed in a human acute lymphoblastic leukemia cell line (NALM6) (*Bertomeu et al., 2018*) ranked *ETFDH* as a top non-essential metabolic gene that clustered with well-established tumor suppressors *TP53* and *RB1* (*Figure 1—figure supplement 1B*). Mining DepMap also revealed that *ETFDH* is essential in only 1 out of 1150 cancer cell lines (*Tsherniak et al., 2017*). This motivated us to examine the role of ETFDH in cancer. *ETFDH* mRNA levels are reduced in patient samples across different cancer types, including colon (*Figure 1—figure supplement 1C*) and breast cancer (*Figure 1—figure supplement 1D*). Immunohistochemistry (IHC) staining confirmed that ETFDH protein levels are significantly reduced in colorectal cancer vs. non-adjacent patient-derived tissues (NAT) (*Figure 1A*).

We next abrogated the *ETFDH* gene using a CRISPR-Cas9 approach in a variety of cancer cell lines, including HCT-116 (human colorectal cancer), NT2197 (murine HER2 +breast cancer), 4T1 (murine triple-negative breast cancer cells), and NALM6 cells (*Figure 1—figure supplement 1E–H*). We also investigated the effects of ETFDH loss in non-transformed normal murine mammary gland (NMuMG) cells (*Figure 1—figure supplement 1I*), a parental immortalized mammary epithelial cell line that was transformed with an oncogenic ErbB2 variant to generate NT2197 cells (*Ursini-Siegel et al., 2008*). Notwithstanding their high baseline proliferation rates, ETFDH ablation significantly increased proliferation across all tested cancer cell lines (*Figure 1—figure supplement 1E–H*). This was not caused by the inadvertent effects of the CRISPR-Cas9, as rescuing ETFDH expression in HCT-116 and NT2197 cells reverted proliferation rates to levels comparable to ETFDH-proficient controls (*Figure 1—figure supplement 1E–F*). In contrast, ETFDH abrogation did not affect proliferation of non-transformed NMuMG cells (*Figure 1—figure supplement 1I*). ETFDH KO HCT-116 cells also formed more colonies in soft agar (surrogate measurement of cellular neoplastic potential *Borowicz et al., 2014*) as compared to empty vector infected controls (WT EV), and ETFDH KO cells in which ETFDH was re-expressed (ETFDH Rescue) (*Figure 1—figure supplement 1J*).

Based on these findings, we tested the impact of ETFDH loss on neoplastic growth in vivo. WT EV, ETFDH KO, or ETFDH Rescue HCT-116 cells were injected orthotopically in the caeca of SCID-BEIGE mice. Loss of ETFDH significantly increased the tumor growth rate and volume at endpoint relative to control and ETFDH Rescue cells (*Figure 1B–C*). ETFDH loss also increased NT2197 tumor growth and

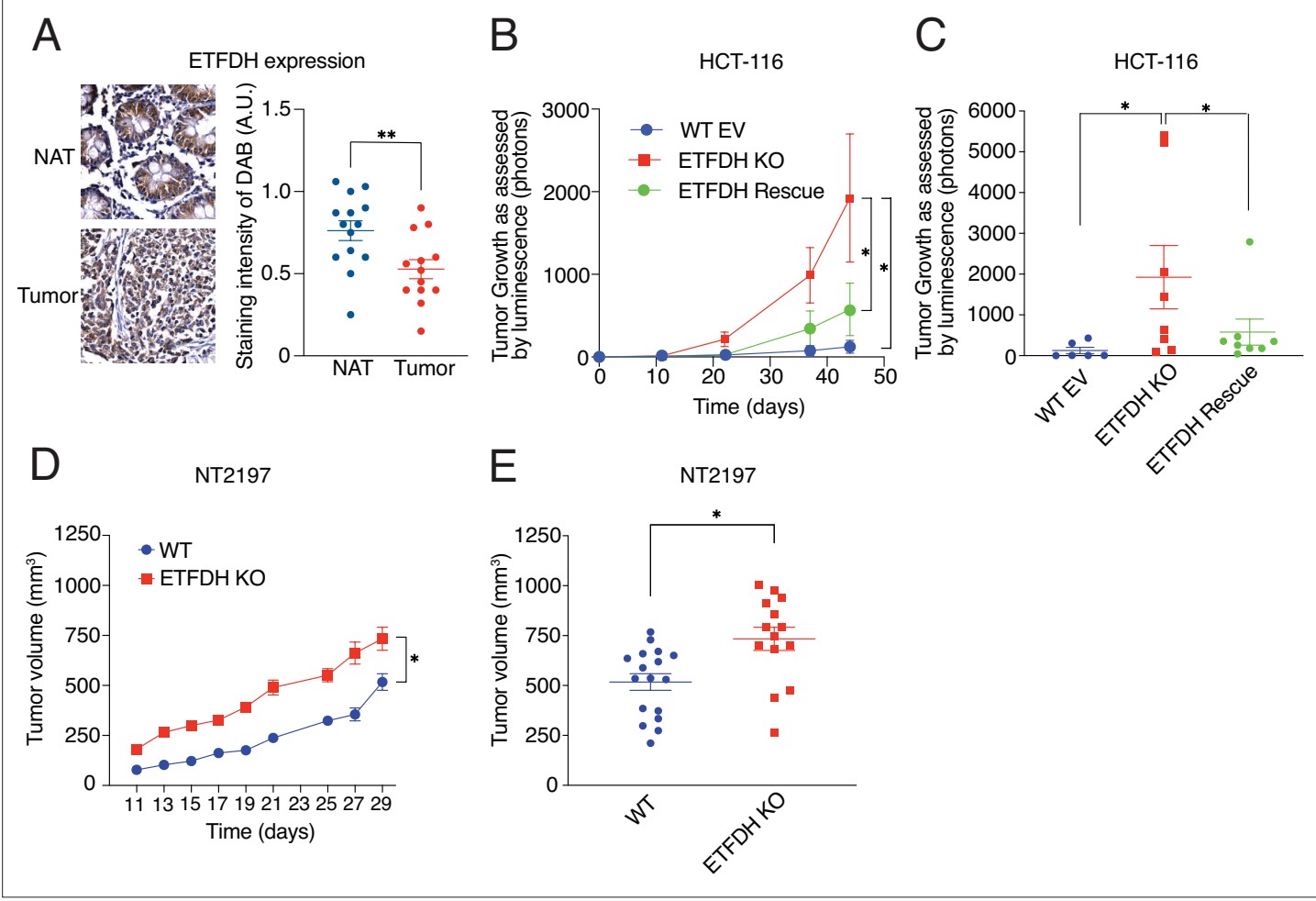

**Figure 1.** Electron transfer flavoprotein dehydrogenase (ETFDH) loss promotes tumor growth. (**A**) ETFDH protein levels in non-adjacent tissue (NAT) or grade 3 cancerous tissue (Tumor) were determined by immunocytochemistry (IHC) of colorectal cancer tissue microarray. Staining intensity of DAB in arbitrary units (A.U.) is shown. Data are presented as means +/- SEM (NAT n=14, Tumor n=13), **$p<0.01$, unpaired Student's $t$ test. (**B–C**) Tumor growth of WT EV, ETFDH KO, and ETFDH Rescue HCT-116 cells following intra-caecal injection (**B**) and endpoint tumor volumes (**C**). Growth was assessed by luminescence (photons). Data are presented as means +/- SEM (WT EV n=6, ETFDH KO n=8, ETFDH Rescue n=8), *$p<0.05$, one-way ANOVA, Tukey's post-hoc test. (**D–E**) Tumor growth of WT and ETFDH KO NT2197 cells following mammary fat-pad injection (**D**) and endpoint tumor volumes (**E**). Growth was assessed using a caliper. Data are presented as means +/- SEM (WT n=17, ETFDH KO n=17), *$p<0.05$, unpaired Student's $t$ test.

The online version of this article includes the following source data and figure supplement(s) for figure 1:

**Figure supplement 1.** *ETFDH* expression is reduced across a broad array of cancer subtypes, whereby *ETFDH* is a non-essential gene in cancer cells.

**Figure supplement 1—source data 1.** PDF files containing original western blots for *Figure 1—figure supplement 1E*, *Figure 1—figure supplement 1F*, *Figure 1—figure supplement 1G*, *Figure 1—figure supplement 1H*, and *Figure 1—figure supplement 1I*, indicating the relevant bands.

**Figure supplement 1—source data 2.** Original files for western blot analysis displayed in *Figure 1—figure supplement 1E*, *Figure 1—figure supplement 1F*, *Figure 1—figure supplement 1G*, *Figure 1—figure supplement 1H*, and *Figure 1—figure supplement 1I*.

the final tumor volume as compared to ETFDH-proficient tumors following orthotopic mammary fat pad injections in SCID-BEIGE mice (*Figure 1D–E*). Altogether, these results show that *ETFDH* is a non-essential gene in cancer that is under-expressed across different cancer types and that its disruption enhances neoplastic growth, while not affecting proliferation of non-transformed cells.

## ETFDH abrogation results in ETF insufficiency that paradoxically increases bioenergetic capacity of cancer cells

In non-cancerous cells, ETFDH transfers electrons from catabolism of fatty acids and amino acids to the ETC (*Figure 2—figure supplement 1A*). Palmitate oxidation (monitored by oxygen consumption)

and $^{13}$C leucine labeling into citrate (citrate m+2) were strongly attenuated by ETFDH abrogation in HCT-116 cells (*Figure 2—figure supplement 1B–C*), which confirmed that ETF plays a role in fatty and amino acid catabolism in malignant cells. Thus, ETFDH loss induces ETF insufficiency in cancer cells that mirrors defects observed in non-malignant tissues. Surprisingly, ETFDH loss in HCT-116 cells increased oxygen consumption and extracellular acidification, relative to control WT EV and ETFDH Rescues (*Figure 2A–B*). ETFDH KO HCT-116 cells produced higher levels of ATP from oxidative phosphorylation (J ATP ox) as compared to ETFDH-proficient cells (*Figure 2C–D*). Consequently, ETFDH loss increased the bioenergetic capacity of HCT-116 cells (*Figure 2E*). Comparable results were observed in NT2197 cells (*Figure 2—figure supplement 1D–H*). Collectively, these findings suggest that although ETF insufficiency limits the ability of cancer cells to utilize lipids and amino acids, it paradoxically upregulates their mitochondrial metabolism and bioenergetic capacity.

Glutamine plays an important role in fueling mitochondrial metabolism and *de novo* nucleotide biosynthesis to support cancer cell proliferation (*DeBerardinis and Cheng, 2010*). ETFDH abrogation increased glutamine uptake and glutamate production in both HCT-116 (*Figure 2F*) and NT2197 (*Figure 2—figure supplement 1I*) cells. ETFDH loss increased $^{13}$C-glutamine incorporation in the forward direction (green) and the reverse direction (purple) of the citric acid cycle (CAC; *Figure 2G-H*, *Figure 2—figure supplement 1J*), suggesting higher glutamine tracing into the CAC, as well as increased reductive glutamine metabolism. Indeed, the steady state levels of several nucleotides were also increased following ETFDH loss (*Figure 2—figure supplement 1K*). To highlight the dependence of glutamine in contributing to the proliferation of cells upon ETFDH abrogation, HCT-116 WT EV, and ETFDH KO cells were grown in the absence or presence of glutamine. Accordingly, ETFDH KO HCT-116 cells exhibited greater sensitivity to glutamine deprivation than control WT EV cells (*Figure 2I*). Collectively, these data demonstrate that ETF insufficiency in cancer cells remodels mitochondrial metabolism and increases glutamine consumption and anaplerosis.

## ETFDH loss induces intracellular accumulation of amino acids, mTOR signaling, and protein synthesis

Consistent with the established role of ETFDH in amino acid catabolism (*Mereis et al., 2021*), and reduction of leucine consumption upon ETFDH abrogation (*Figure 2—figure supplement 1C*), steady-state levels of most intracellular amino acids were increased in ETFDH KO vs. WT HCT-116 and NT2197 cells (*Figure 3A*). Moreover, ETFDH-deficient HCT-116 cells exhibited higher rates of protein synthesis relative to ETFDH-proficient cells (*Figure 3B*). Amino acids activate the mechanistic target of rapamycin (mTOR) which stimulates protein synthesis (*Saxton and Sabatini, 2017*). Indeed, mTORC1 signaling was higher in ETFDH-deficient than ETFDH-proficient HCT-116 and NT2197 cells as illustrated by increased phosphorylation of ribosomal protein S6 kinases 1 and 2 (S6K1; T389) and their substrate ribosomal protein S6 (S6; S240/244) (*Figure 3C*). Re-expression of ETFDH in ETFDH KO cells resulted in normalization of mTORC1 signaling to the levels observed in WT EV cells (*Figure 3D*). mTORC2 signaling was also augmented upon ETFDH abrogation in HCT-116 and NALM6 cells, as evidenced by increased AKT phosphorylation (S473) (*Figure 3—figure supplement 1A–B*). Despite bioenergetic rewiring, ETFDH loss did not exert a major effect on the activity of adenosine monophosphate-activated protein kinase (AMPK) as monitored by its phosphorylation (T172) or phosphorylation of its substrate acetyl-CoA carboxylase (ACC; S79) (*Figure 3—figure supplement 1A–B*).

Consistent with mTOR hyperactivation, ETFDH KO HCT-116 cells were more sensitive to the active site mTOR inhibitor Torin1 (*Thoreen et al., 2009*) as compared to their ETFDH-proficient counterparts (*Figure 3—figure supplement 1C*). To further determine which mTOR complex is responsible for mediating the hyper-proliferative phenotype triggered by ETFDH loss, we selectively inactivated mTORC1 or mTORC2 by depleting RAPTOR or RICTOR, respectively, using small hairpin RNA (shRNA) in WT and ETFDH KO HCT-116 cells (*Figure 3—figure supplement 1D*). Depletion of RAPTOR, but not RICTOR, strongly attenuated the proliferation of ETFDH KO cells (*Figure 3—figure supplement 1E–F*). Moreover, the anti-proliferative effects of bisteric mTORC1-specific inhibitor BiS-35x (*Lee et al., 2021*) were stronger in ETFDH KO vs. ETFDH Rescue HCT-116 cells (*Figure 3—figure supplement 1G*). Altogether, these data suggest that mTORC1, but not mTORC2, acts as a major mediator of hyper-proliferative phenotype triggered by ETFDH loss in cancer cells. Accordingly, the effects of amino acid deprivation on phospho-S6 (S240/244) and phospho-4E-BP1 (S65) were attenuated in ETFDH KO vs. WT control cells, which is consistent with the intracellular accumulation of amino acids

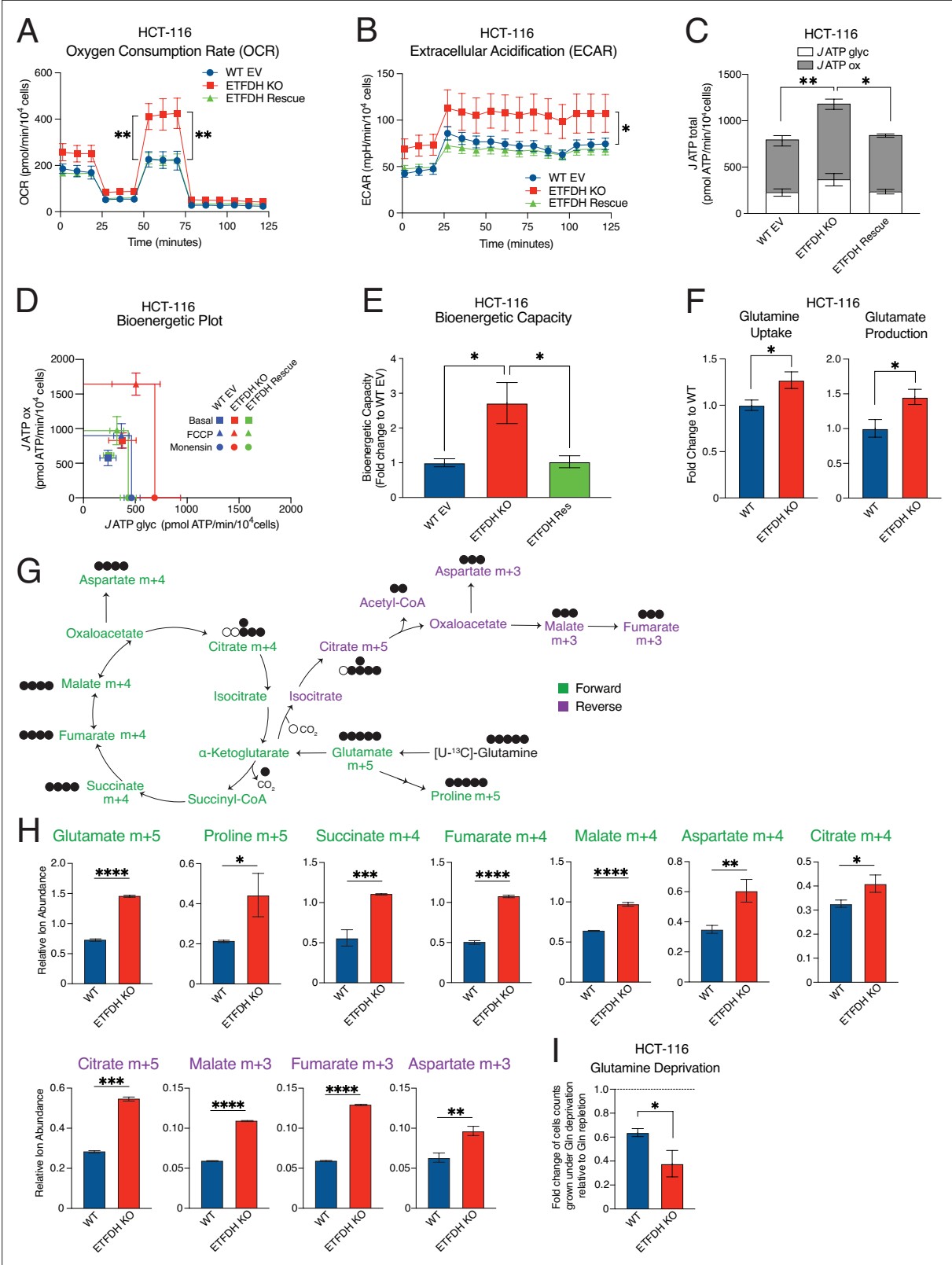

**Figure 2.** Absence of electron transfer flavoprotein dehydrogenase (ETFDH) promotes mitochondrial metabolism. (**A–B**) Oxygen consumption (**A**) and extracellular acidification (**B**) in WT EV, ETFDH KO, and ETFDH Rescue HCT-116 cells were determined using a Seahorse bioanalyzer. Data are normalized to cell count and presented as means +/- SD (n=4), *$p<0.05$, **$p<0.01$, one-way ANOVA, Tukey's post-hoc test.(**C**) Basal $J$ ATP calculations from WT EV, ETFDH KO, and ETFDH Rescue HCT-116 cells. $J$ ATP ox represents ATP production from oxidative phosphorylation, while $J$ ATP glyc is

*Figure 2 continued on next page*

*Figure 2 continued*

ATP production from glycolysis. Comparison between *J* ATP ox (gray bars; top) and *J* ATP glyc (white bars; bottom) is shown. Data are presented as means +/- SD (n=4), *p<0.05, **p<0.01, one-way ANOVA, Tukey's post-hoc test. (**D–E**) Bioenergetic plot for Basal, FCCP, and Monensin *J* ATP fluxes (**D**) and bioenergetic capacity (**E**) of WT EV, ETFDH KO, and ETFDH Rescue HCT-116 cells. Data are presented as means +/- SD (n=4), *p<0.05, one-way ANOVA, Tukey's post-hoc test. (**F**) Glutamine uptake and glutamate production in WT and ETFDH KO HCT-116 cells. Results are shown as fold changes of ETFDH KO cells relative to WT cells. Data are presented as means +/- SD (n=3), *p<0.05, paired Student's *t* test. (**G**) Schematic of ¹³C-glutamine tracing throughout the citric acid cycle (CAC). Isotopomers labeled in green depict ¹³C-glutamine tracing in the forward direction of the CAC, while those in purple represent reverse tracing. (**H**) Relative abundance of ¹³C-labeled metabolites in the forward (green) and reverse (purple) directions from WT (blue) and ETFDH KO (red) NT2197 cells. Data are presented as means +/- SD (n=3), *p<0.05, **p<0.01, ***p<0.001, ****p<0.0001, paired Student's *t* test. (**I**) Proliferation of WT and ETFDH KO HCT-116 cells grown in the absence or presence of glutamine for 48 hr. Results are shown as fold changes of cell counts under glutamine deprivation relative to those under glutamine repleted conditions. Data are presented as means +/- SD (n=3), *p<0.05, paired Student's *t* test.

The online version of this article includes the following figure supplement(s) for figure 2:

**Figure supplement 1.** Electron transfer flavoprotein dehydrogenase (ETFDH) loss reprograms mitochondrial bioenergetics.

upon ETFDH disruption (*Figure 3E–G*). In turn, serum stimulation (Stim) resulted in comparable induction in phosphorylation of S6 (S240/244) and 4E-BP1 (S65) in ETFDH-deficient and proficient cells (*Figure 3E*). Collectively, these findings suggest that ETFDH loss induces mTORC1 signaling at least in part by increasing intracellular amino acid levels.

Amino acid deprivation leads to general control nonderepressible 2 (GCN2)-dependent phosphorylation of the alpha subunit of eIF2 and subsequent induction of activating transcription factor 4 (ATF4) synthesis during the integrated stress response (ISR) (*Harding et al., 2000*). Consistent with an increase in intracellular amino acid, phospho-eIF2α (S51) levels appeared to be modestly reduced in ETFDH-deficient vs. proficient HCT-116 cells under both serum starvation and amino acid depletion (*Figure 3—figure supplement 1H*). However, the ETFDH status in the cells did not affect ATF4 levels at baseline or upon amino acid depletion (*Figure 3—figure supplement 1H*). Altogether, these data demonstrate that ETFDH loss induces mTORC1 signaling and protein synthesis, which appears to be at least in part driven by intracellular accumulation of amino acids.

## ETFDH loss diminishes 4E-BP1, but not 4E-BP2 protein levels

mTORC1 positively controls cell proliferation through the eukaryotic translation initiation factor 4E-binding proteins (4E-BPs; 4E-BP1-3 in mammals) (*Dowling et al., 2010*). Of note, 4E-BP3 is expressed in a tissue-restricted manner and does not appear to play a major role in regulating proliferation (*Dowling et al., 2010*). mTORC1 phosphorylates 4E-BPs to promote eukaryotic translation initiation factor 4 F (eIF4F) complex assembly by dissociating 4E-BPs from eIF4E, thereby stimulating cap-dependent translation initiation (*Gingras et al., 1999*). Consistent with the increased mTORC1 activity, ETFDH-deficient cells exhibited increased phosphorylation of 4E-BP1 (S65), which was rescued upon re-expression of ETFDH in HCT-116 and NT2197 cells (*Figure 4A*). Unexpectedly, we observed that ETFDH loss coincided with a dramatic reduction in total 4E-BP1 protein levels in HCT-116 (*Figure 4A*), NT2197 (*Figure 4A*), NALM6 (*Figure 4—figure supplement 1A*), and 4T1 cells (*Figure 4—figure supplement 1B*). We also treated WT and ETFDH KO HCT-116 cell lysates with $\lambda$-phosphatase to exclude potential confounding effects of different 4E-BP1 phosphorylation states, which confirmed that disruption of ETFDH is paralleled by reduction in 4E-BP1 levels (*Figure 4B*). Notably, loss of ETFDH did not exert a major effect on 4E-BP2 protein abundance in HCT-116 (*Figure 4A*), NT2197 (*Figure 4A*), or 4T1 cells (*Figure 4—figure supplement 1B*). In addition, 4E-BP1, but not 4E-BP2 protein levels were reduced in ETFDH-deficient as compared to WT HCT-116 tumors (*Figure 4C–E*). Strikingly, loss of ETFDH in non-transformed NMuMG cells did not exert a major effect on mTORC1 signaling (*Figure 4—figure supplement 1C*) and failed to alter 4E-BP1 protein levels (*Figure 4—figure supplement 1C*). Collectively, these findings indicate that ETF insufficiency is paralleled by a selective decrease in 4E-BP1 protein levels in cancer, but not non-transformed cells.

## Decrease in 4E-BP1 protein levels underpins the bioenergetic rewiring induced by ETF insufficiency in cancer cells

Considering that in mammals 4E-BPs play a major role in mediating the effects of mTORC1 on proliferation, neoplastic growth, mitochondrial functions, and protein synthesis (*Dowling et al., 2010*;

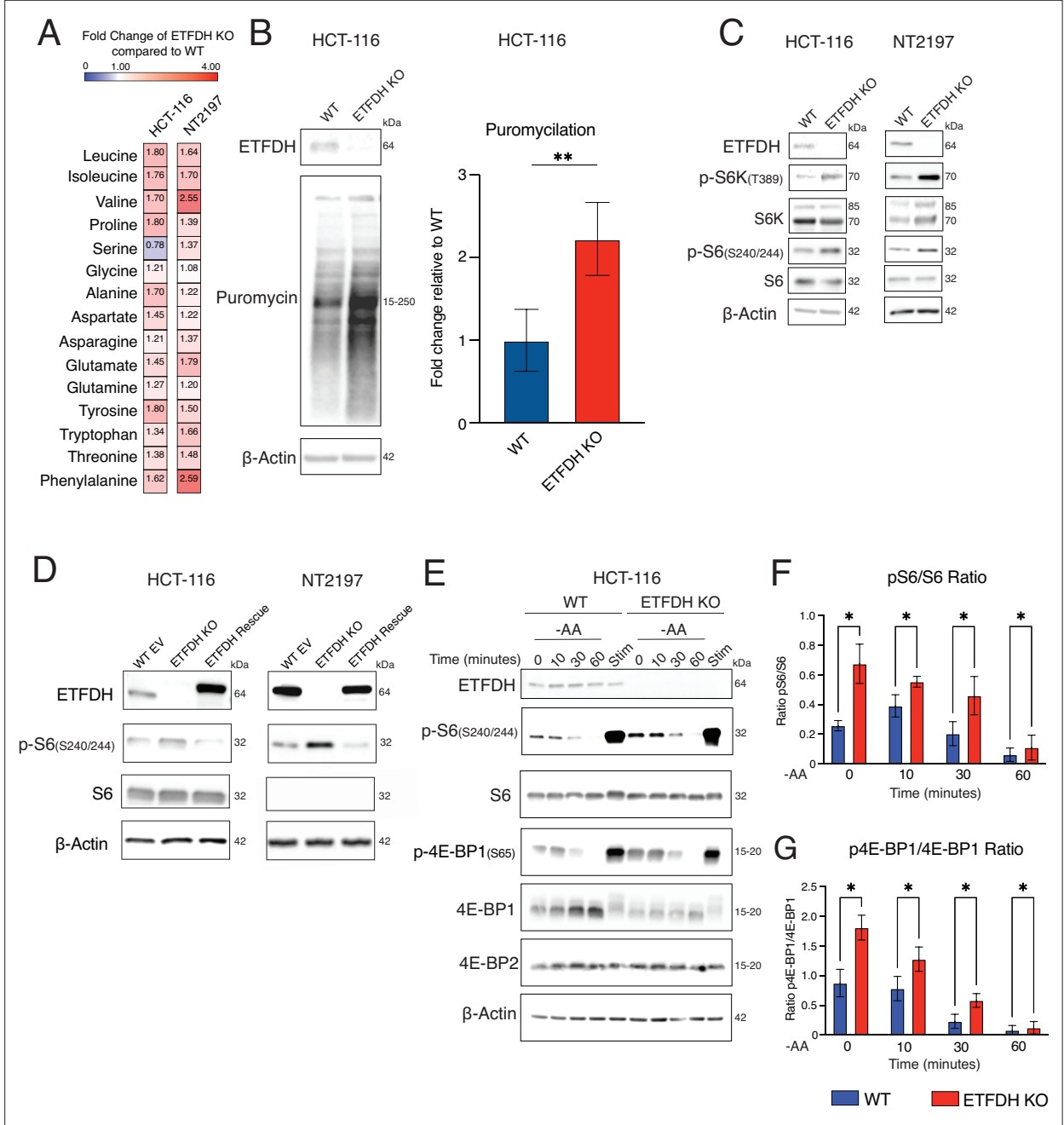

**Figure 3.** Electron transfer flavoprotein dehydrogenase (ETFDH) loss bolsters mTORC1 signaling and downregulates 4E-BP1 protein levels. (**A**) Amino acid profile in WT and ETFDH KO HCT-116 and NT2197 cells. Data are presented as fold change in amino acid levels in ETFDH KO compared to WT cells (n=3). (**B**) Puromycin incorporation assays in WT and ETFDH KO HCT-116 cells. Levels and phosphorylation status of indicated proteins were determined via western blotting using indicated antibodies. β-Actin was used as a loading control (representative blots of n=3). Quantification of puromycin incorporation is presented as fold change in ETFDH KO vs. WT cells. (**C**) Levels and phosphorylation status of indicated proteins in WT or ETFDH KO HCT-116 and NT2197 cells were determined by western blotting. β-Actin served as a loading control (representative blots of n=3). (**D**) Levels and phosphorylation status of indicated proteins in WT EV, ETFDH KO, or ETFDH Rescue HCT-116 and NT2197 cells were determined by western blotting. β-Actin was used as a loading control (representative blots of n=3). (**E**) Total and phosphoprotein levels in WT or ETFDH KO HCT-116 cells were determined by western blotting using indicated proteins. Cells were serum starved overnight, then depleted of amino acids for indicated time points (0, 10, 30, 60 min), or stimulated with FBS for 4 hr (Stim). β-Actin was used as loading control (representative blots of n=3). (**F**) Quantification of pS6(S240/244)/S6 ratio from data presented in panel E. Data are presented as means +/- SD (n=3), *$p<0.05$, one-way ANOVA, Tukey's post-hoc test. (**G**) Quantification of p4E-BP1(S65)/4E-BP1 ratio from data presented in panel E. Data are presented as means +/- SD (n=3), *$p<0.05$, one-way ANOVA, Tukey's post-hoc test.

*Figure 3 continued on next page*

*Figure 3 continued*

The online version of this article includes the following source data and figure supplement(s) for figure 3:

**Source data 1.** PDF files containing original western blots for *Figure 3B and C*, *Figure 3D , and E*, indicating the relevant bands.

**Source data 2.** Original files for western blot analysis displayed in *Figure 3B and C*, *Figure 3D , and E*.

**Figure supplement 1.** Electron transfer flavoprotein dehydrogenase (ETFDH) loss increases mTORC1 signaling and dependency.

**Figure supplement 1—source data 1.** PDF files containing original western blots for *Figure 3—figure supplement 1A*, *Figure 3—figure supplement 1B*, *Figure 3—figure supplement 1D*, and *Figure 3—figure supplement 1H*, indicating the relevant bands.

**Figure supplement 1—source data 2.** Original files for western blot analysis displayed in *Figure 3—figure supplement 1A*, *Figure 3—figure supplement 1B*, *Figure 3—figure supplement 1D*, and *Figure 3—figure supplement 1H*.

*Hulea et al., 2018*; *Morita et al., 2013*), we next investigated whether the observed decrease in 4E-BP1 levels contributes to increased proliferation and tumor growth caused by ETFDH abrogation. To this end, we overexpressed 4E-BP1 protein in ETFDH-deficient NT2197 cells by infecting them with a retrovirus wherein *EIF4EBP1* expression was driven by an exogenous promoter (*Dowling et al., 2010*). Importantly, 4E-BP1 overexpression in ETFDH KO NT2197 cells decreased eIF4F complex assembly, as evidenced by m$^7$-GTP pulldown (*Figure 4F*). 4E-BP1 overexpression also reduced oxygen consumption in ETFDH KO NT2197 cells, relative to empty vector-infected cells (*Figure 4G*). Moreover, restoring 4E-BP1 protein levels decreased cell proliferation (*Figure 4—figure supplement 1D*) and anchorage-independent growth (*Figure 4—figure supplement 1E*) of ETFDH KO HCT-116 cells. Lastly, 4E-BP1 overexpression decreased growth and the final volume of tumors formed by NT2197 ETFDH KO cells in SCID-BEIGE mice (*Figure 4H–I*). Collectively, these data indicate that decreased 4E-BP1 protein levels mediate the pro-neoplastic effects of ETF insufficiency.

The mTORC1/4E-BP axis stimulates mitochondrial biogenesis and metabolism by altering the synthesis of nuclear-encoded proteins with mitochondrial functions, including mitochondrial transcription factor A (TFAM) (*Morita et al., 2013*; *Morita et al., 2015*). ETFDH loss increased mitochondrial DNA levels (*Figure 4J*) and mitochondrial mass (*Figure 4—figure supplement 1F*), which was paralleled by TFAM induction (*Figure 4K*). Collectively, these data show that the loss of 4E-BP1 plays a prominent role in mediating the effects of ETF insufficiency on bioenergetic reprogramming of cancer cells.

## ETFDH loss decreases *EIF4EBP1* transcription via BCL-6

We next set out to determine the level(s) at which 4E-BP1 protein levels are downregulated in ETF-insufficient cancer cells. ETFDH loss did not have a major effect on 4E-BP1 protein synthesis or turnover (*Figure 4—figure supplement 1G–H*), but it decreased the *EIF4EBP1* mRNA levels in ETFDH-deficient vs. ETFDH-proficient HCT-116 (*Figure 4L*) and NT2197 cells (*Figure 4M*). *EIF4EBP1* and *ETFDH* mRNA levels were also positively correlated in tumors isolated from colorectal adenocarcinoma patients (*Figure 4—figure supplement 1I*). In turn, consistent with no apparent differences in 4E-BP2 protein levels, the levels of *EIF4EBP2* mRNA were unaffected by the ETFDH status in both cell lines (*Figure 4L–M*). Notably, the observed decrease in *EIF4EBP1* mRNA levels was not caused by the effects of ETFDH loss on mRNA stability (*Figure 4—figure supplement 1J*). Based on these data, we concluded that ETFDH abrogation is paralleled by suppression of *EIF4EBP1* transcription and sought to identify transcription factors that may mediate the effects of ETFDH abrogation on suppression of *EIF4EBP1* expression. ATF4, Snail, and Slug have been shown to stimulate transcription of *EIF4EBP1*, but not *EIF4EBP2* (*Yamaguchi et al., 2008*; *Wang et al., 2017*). However, ATF4, Snail, and Slug proteins remain unchanged upon ETFDH disruption (*Figure 3—figure supplement 1H*, *Figure 5—figure supplement 1A*). We also re-analyzed publicly available RNA-seq datasets, which revealed that *EIF4EBP1,* but not *EIF4EBP2* expression was increased following STAT1 abrogation in MT4788 breast cancer cells (*Totten et al., 2021*). Indeed, we observed that MT4788 STAT1 KO cells have increased 4E-BP1 protein levels relative to parental cells (*Figure 5—figure supplement 1B*). Accordingly, ETFDH KO cells displayed increased STAT1 and downregulated 4E-BP1 protein levels (*Figure 5—figure supplement 1C*). To determine the impact of STAT1 on *EIF4EBP1* or *EIF4EBP2* promoter occupancy, we performed chromatin immunoprecipitation qPCR (ChIP-qPCR) assay in control (WT EV), ETFDH KO, or ETFDH rescue HCT-116 cells (*Figure 5—figure supplement 1D–E*). Binding of STAT1 was, however, comparable to IgG across all tested cell lines, thereby suggesting

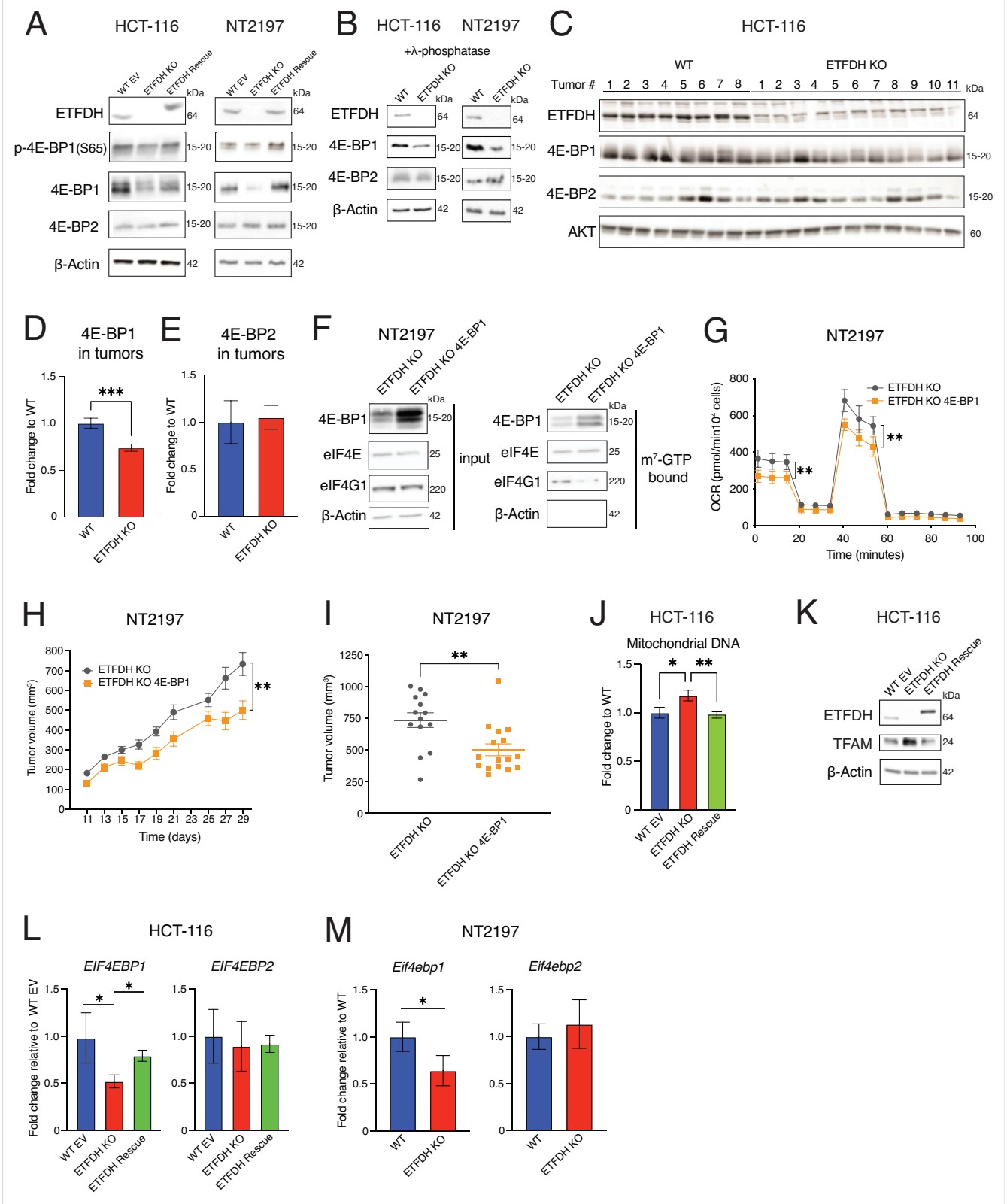

**Figure 4.** Repression of *EIF4EBP1* transcription mediates the effects of electron transfer flavoprotein dehydrogenase (ETFDH) loss on cancer bioenergetics and tumor growth. (**A**) Levels and phosphorylation status of indicated proteins in WT EV, ETFDH KO, or ETFDH Rescue HCT-116 and NT2197 cells were determined by western blotting. β-Actin was used as a loading control (representative blots of n=3). (**B**) Levels of indicated protein in λ-phosphatase treated WT or ETFDH KO HCT-116 and NT2197 cell lysates were monitored by western blotting. β-Actin was used as a loading

*Figure 4 continued on next page*

*Figure 4 continued*

control (representative blots of n=3). (**C**) Indicated protein levels in WT or ETFDH KO HCT-116 tumors were determined by western blotting. AKT was used as a loading control (WT n=8, ETFDH KO n=11). (**D–E**) Quantification of 4E-BP1 (**D**) and 4E-BP2 (**E**) from tumors described in panel C. Data are presented as means +/- SEM (WT n=8, ETFDH KO n=11), ***$p<0.001$, unpaired Student's *t* test. (**F**) $m^7$GDP pulldown assay in ETFDH KO and ETFDH KO overexpressing 4E-BP1 (ETFDH KO 4E-BP1) NT2197 cells. Specified protein levels in $m^7$GDP pulldown or input were determined by western blot. β-Actin was used as loading control (input) and to exclude contamination in the pulldown material ($m^7$GTP bound) (representative blots of n=3). (**G**) Oxygen consumption of ETFDH KO and ETFDH KO 4E-BP1 NT2197 cells. Data are normalized to cell count and presented as means +/- SD (n=5), **$p<0.01$, Student's *t* test. (**H–I**) Tumor growth of ETFDH KO and ETFDH KO 4E-BP1 NT2197 cells following mammary fat-pad injection (**H**) and endpoint tumor volumes (**I**). Growth was assessed using caliper measurements. ETFDH KO measurements are the same as in *Figure 1D*, as these growth curves were obtained in the same experiment. Data are presented as means +/- SEM (ETFDH KO n=17, ETFDH KO 4E-BP1 n=18), **$p<0.01$, unpaired Student's *t* test. (**J**) Mitochondrial DNA in WT EV, ETFDH KO, and ETFDH Rescue HCT-116 cells was quantified by qPCR. Mitochondrial DNA (mtDNA) content was normalized to genomic DNA (gDNA). Data are presented as fold change relative to WT EV cells +/- SD (n=3), *$p<0.05$, **$p<0.01$, one-way ANOVA, Tukey's post-hoc test. (**K**) Levels of indicated proteins in WT EV, ETFDH KO, and ETFDH Rescue HCT-116 cells were assessed by western blotting. β-Actin was used as a loading control (representative blots of n=3). (**L**) *EIF4EBP1* and *EIF4EBP2* mRNA levels in WT EV, ETFDH KO, and ETFDH Rescue HCT-116 cells were determined by RT-qPCR. *PP1A* was used as a housekeeping gene. Data are presented as fold change of *EIF4EBP1/PP1A* and *EIF4EBP2/PP1A* ratios relative to WT EV cells (set to 1)+/-SD (n=3), *$p<0.05$, one-way ANOVA, Tukey's post-hoc test. (**M**) *Eif4ebp1* and *Eif4ebp2* mRNA abundance in WT and ETFDH NT2197 KO cells (murine cell line) was determined by RT-qPCR. *Actb* was used as a housekeeping gene. Data are presented as fold change in *Eif4ebp1/Actb* and *Eif4ebp2/Actb* ratios relative to WT cells +/- SD (n=3), *$p<0.05$, paired Student's *t* test.

The online version of this article includes the following source data and figure supplement(s) for figure 4:

**Source data 1.** PDF files containing original western blots for *Figure 4A and B*, *Figure 4C and F*, and *Figure 4K*, indicating the relevant bands.

**Source data 2.** Original files for western blot analysis displayed in *Figure 4A and B*, *Figure 4C and F*, and *Figure 4K*.

**Figure supplement 1.** *Electron transfer flavoprotein dehydrogenase (ETFDH)* loss induces *EIF4EBP1* transcription.

**Figure supplement 1—source data 1.** PDF files containing original western blots for *Figure 4—figure supplement 1A*, *Figure 4—figure supplement 1B*, *Figure 4—figure supplement 1C*, and *Figure 4—figure supplement 1G*, indicating the relevant bands.

**Figure supplement 1—source data 2.** Original files for western blot analysis displayed in *Figure 4—figure supplement 1A*, *Figure 4—figure supplement 1B*, *Figure 4—figure supplement 1C*, and *Figure 4—figure supplement 1G*.

that it is unlikely that STAT1 directly suppresses *EIF4EBP1* transcription in the context of ETFDH loss (*Figure 5—figure supplement 1D–E*). Furthermore, reactive oxygen species (ROS) can also influence redox-responsive transcription (*Hayes et al., 2020*). Although loss of ETFDH induced ROS levels (*Figure 5—figure supplement 1F*), 4E-BP1 protein abundance was not altered following treatment with the antioxidant N-acetyl-cysteine (NAC) (*Figure 5—figure supplement 1G*).

We next investigated the potential role of B-cell lymphoma 6 (BCL-6) (*Nakayamada et al., 2014*; *Liu et al., 2022*; *Madapura et al., 2017*), which was previously suggested to suppress *EIF4EBP1* transcription (*Ramachandran et al., 2024*). Notably, *ETFDH* abrogation in HCT-116 and NT2197 cells resulted in a marked increase in BCL-6 protein levels, compared to ETFDH-proficient control cells (*Figure 5A*). These changes in BCL-6 protein levels were mTOR-sensitive (*Figure 5B*) and not accompanied by the alterations in *BCL6* mRNA abundance (*Figure 5C*), which is consistent with recently reported mTOR-dependent translational activation of BCL-6 (*Yi et al., 2017*; *Xu et al., 2017*; *Raybuck et al., 2018*). Indeed, *BCL6* mRNA translation was increased in HCT-116 ETFDH KO relative to WT cells (*Figure 5D*). Taken together, these data point out that ETFDH disruption induces BCL-6 protein synthesis in an mTOR-dependent manner. ETF insufficiency increased BCL-6 binding to the *EIF4EBP1*, but not the *EIF4EBP2* promoter (*Figure 5E–F*). BCL-6 silencing in NT2197 and 4T1 ETFDH KO cells increased 4E-BP1 protein levels (*Figure 5G–H*). Depletion of BCL-6 also attenuated proliferation of 4T1 ETFDH KO cells (*Figure 5I*). Collectively, these findings show that BCL-6 mediates the effects of ETF insufficiency on reduction in *EIF4EBP1* expression and increased proliferation of cancer cells.

## ETFDH acts as a haploinsufficient tumor suppressor

Data available in cBioPortal from 2683 samples across cancer types (*Aaltonen et al., 2020*) indicate that missense or truncating mutations in ETFDH are rare (0.3%) (*Figure 6A*). This suggests that ETFDH is not lost, but rather downregulated in human cancers. We observed that DNA methylation inhibitors, 5-azacytidine (5-aza) and 5-aza-2'-deoxycytidine (5-aza-2-deoxyC) increased ETFDH protein levels in HCT-116 cells (*Figure 6B*), thus suggesting that one of the potential mechanisms of downregulated *ETFDH* expression in neoplasia may stem from increased DNA methylation. Based on these observations, we developed a model in murine NT2197 cells that more closely reflects

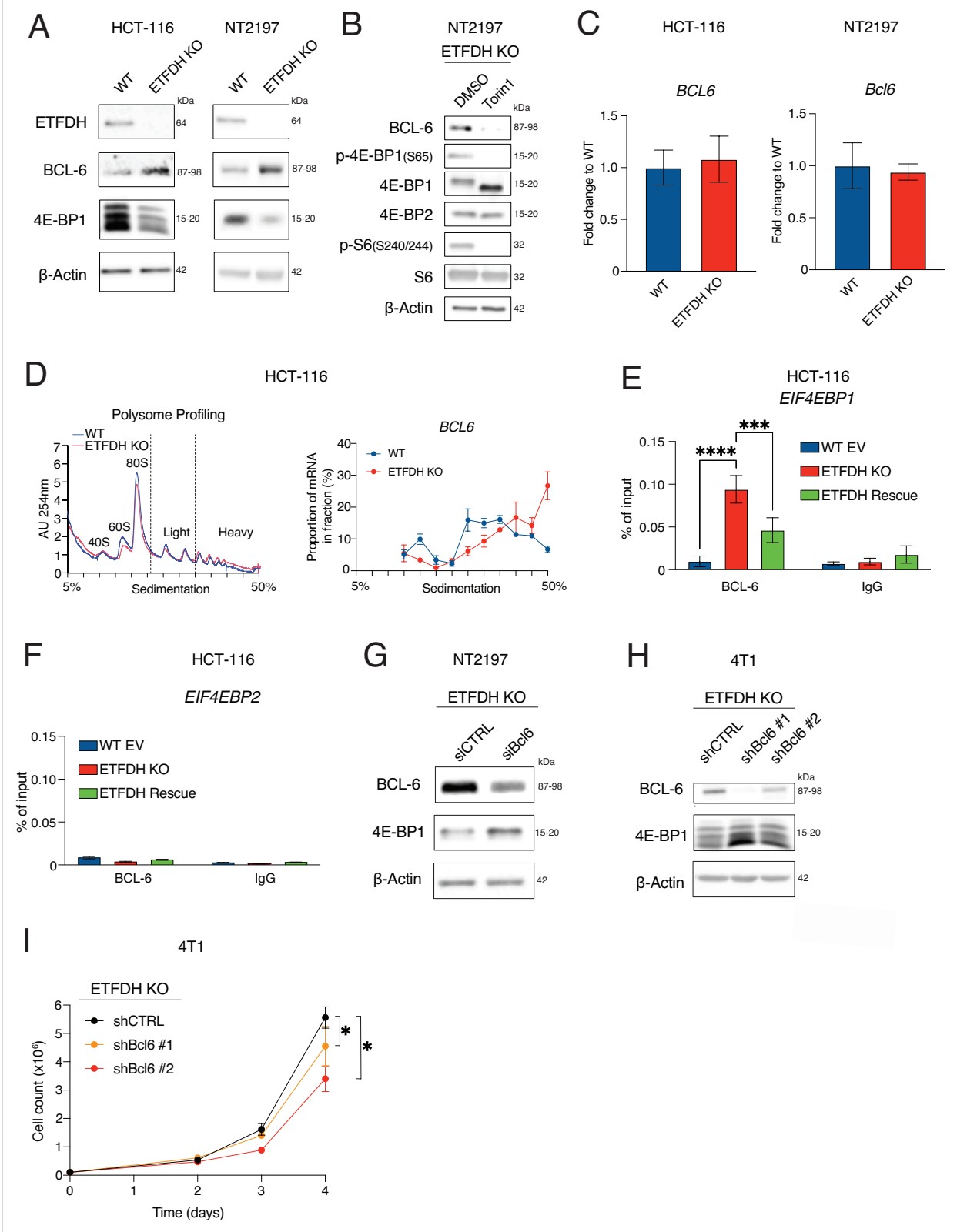

**Figure 5.** Loss of electron transfer flavoprotein dehydrogenase (ETFDH) increases BCL-6 levels leading to BCL-6-dependent inhibition of *EIF4EBP1* transcription. (**A**) Levels of indicated proteins in WT or ETFDH KO HCT-116 and NT2197 cells were determined by western blotting. β-Actin was used as a loading control (representative blots of n=3). (**B**) Levels of indicated proteins in ETFDH KO NT2197 cells treated with DMSO or Torin1 (250 nM) for 4 hr were determined by western blotting. β-Actin was used as a loading control (representative blots of n=3). (**C**) *BCL6/Bcl6* mRNA abundance in WT

*Figure 5 continued on next page*

*Figure 5 continued*

and ETFDH KO HCT-116 and NT2197 cells was determined by RT-qPCR. *PP1A* mRNA was used as a control for HCT-116 experiments, while *Actb* mRNA was used as a control for NT2197 experiments. Data are presented as fold change in *EIF4EBP1/PP1A* and *EIF4EBP2/PP1A* (HCT-116) and *Eif4ebp1/ Actb* and *Eif4ebp2/Actb* (NT2197) ratios relative to WT cells +/- SD (n=3), paired Student's *t* test. (**D**) Polysome profiling of WT and ETFDH HCT-116 KO cells. The absorbance at 254 nm (AU 254 nm) was used to monitor distribution of the 40S-, 60S- ribosomal subunits, monosomes (80 S), and polysomes across the gradient. *BCL6* mRNA abundance from polysome gradient fractions was obtained using RT-qPCR. Data are shown as a mean percentage of mRNA in each fraction relative to cumulative corresponding mRNA amount across the whole gradient +/- SD (n=3). (**E–F**) Binding events of BCL-6 to the promoters of *EIF4EBP1* and *EIF4EBP2* were determined by ChIP-qPCR. IgG was used as a negative control. Data are presented as % of input (n=3), *p<0.05, two-way ANOVA, Dunnett's post-hoc test. (**G**) The levels of indicated proteins in ETFDH KO NT2197 cells transfected with siRNA targeting BCL-6 (siBCL-6) or control, scrambled siRNA (siCTRL) were determined by western blotting. β-Actin was used as loading control (representative blots of n=2). (**H**) Levels of indicated proteins in ETFDH KO 4T1 cells infected with shRNAs targeting BCL-6 (shBCL-6 #1 or shBCL-6#2) or control, scrambled shRNA (shCTRL) were monitored by western blotting. β-Actin served as loading control (representative blots of n=3). (**I**) Proliferation of ETFDH KO shCTRL, shBCL-6 #1, or shBCL-6#2 4T1 cells. Data are presented as cell count means (n=3), *p<0.05, one-way ANOVA, Dunnett's post-hoc test.

The online version of this article includes the following source data and figure supplement(s) for figure 5:

**Source data 1.** PDF files containing original western blots for *Figure 5A and B*, *Figure 5G , and H*, indicating the relevant bands.

**Source data 2.** Original files for western blot analysis displayed in *Figure 5A and B*, *Figure 5G , and H*.

**Figure supplement 1.** The effects of ETFDH loss on *EIF4EBP1* transcription are mediated by BCL-6 but not STAT1, Snail, or Slug.

**Figure supplement 1—source data 1.** PDF files containing original western blots for *Figure 5—figure supplement 1A*, *Figure 5—figure supplement 1B*, *Figure 5—figure supplement 1C*, and *Figure 5—figure supplement 1G*, indicating the relevant bands.

**Figure supplement 1—source data 2.** Original files for western blot analysis displayed in *Figure 5—figure supplement 1A*, *Figure 5—figure supplement 1B*, *Figure 5—figure supplement 1C*, and *Figure 5—figure supplement 1G*.

---

ETFDH downregulation, but not a complete loss, in human tumors using a method based on targeted poly(A) track insertion within endogenous genes (*Powell et al., 2021*). The length of the poly(A) track directly correlates with the reduction in amount of protein produced, allowing for the investigation of gene-dosage effects on ensuing phenotypes (*Powell et al., 2021*; *Arthur et al., 2015*). Accordingly, mutants were designed by inserting poly(A) tracks of 12 and 18 adenosines [12 A, equivalent to four consecutive AAA (Lys) codons $(AAA)_4$; 18 A, equivalent to six consecutive AAA (Lys) codons $(AAA)_6$], as well as a control track (CTRL) [six consecutive lysine AAG codons $(AAG)_6$] (*Figure 6—figure supplement 1A*, see methods). Based on an Alphafold (version 2) rendered structure (*Jumper et al., 2021*; *Varadi et al., 2022*) of murine ETFDH, we chose residue 38 in an unstructured loop as an insertion site (*Figure 6—figure supplement 1B*). Of note, $(AAA)_6$ and $(AAG)_6$ stretches are translated in the equal number of Lys residues, whereby (AAA), but not (AAG), stretches decrease ETFDH protein levels in NT2197 cells (*Figure 6C*). $(AAG)_6$ construct was thus used as a control to mitigate potential inadvertent effects of the insertion of poly-Lys on the ETFDH function. $(AAA)_4$ and $(AAA)_6$ stretches progressively reduced ETFDH protein abundance as compared to constructs harboring the $(AAG)_6$ track (*Figure 6C*). The reduction in ETFDH protein levels was mirrored by a gene-dose-dependent increase in mTORC1 signaling, BCL-6 expression, and reduction in 4E-BP1, but not 4E-BP2 protein levels (*Figure 6C*). Accordingly, *Eif4ebp1* mRNA levels were reduced in an *ETFDH*-dose-dependent manner, while *Eif4ebp2* mRNA levels remained unaffected (*Figure 6D*). Moreover, reduction in ETFDH protein levels correlated with increased mtDNA content (*Figure 6E*), oxygen consumption (*Figure 6F*), ATP production from OXPHOS (JATPox) (*Figure 6G*), bioenergetic capacity (*Figure 6H*), and cell proliferation (*Figure 6I*). Finally, ETFDH KO NT2197 tumors carrying $(AAA)_6$ insertions exhibit increased tumor growth and larger final tumor volumes relative to control tumors expressing $(AAG)_6$ insertions in SCID-BEIGE mice (*Figure 6J–K*). Altogether, these findings show that ETFDH gene dosage influences neoplastic growth, thus suggesting that ETFDH may act as a haploinsufficient tumor suppressor.

Lastly, we determined whether the catalytic activity of ETFDH is required for its tumor suppressive activity. To achieve this, we re-expressed WT or ETFDH mutant (Y304A, G306E) with disrupted catalytic activity (*Herrero Martín et al., 2024*) in HCT-116 ETFDH KO cells. Notwithstanding that both ETFDH variants were expressed to comparable levels, in contrast to WT, ETFDH (Y304A, G306E) mutant failed to suppress mTORC1 signaling and decrease 4E-BP1 levels (*Figure 6—figure supplement 1C*), reduce proliferation (*Figure 6—figure supplement 1D*) or perturb the bioenergetic profiles of HCT-116 ETFDH KO cells (*Figure 6—figure supplement 1E–F*). Taken together, these data suggest the catalytic activity of ETFDH is required for its tumor suppressive actions.

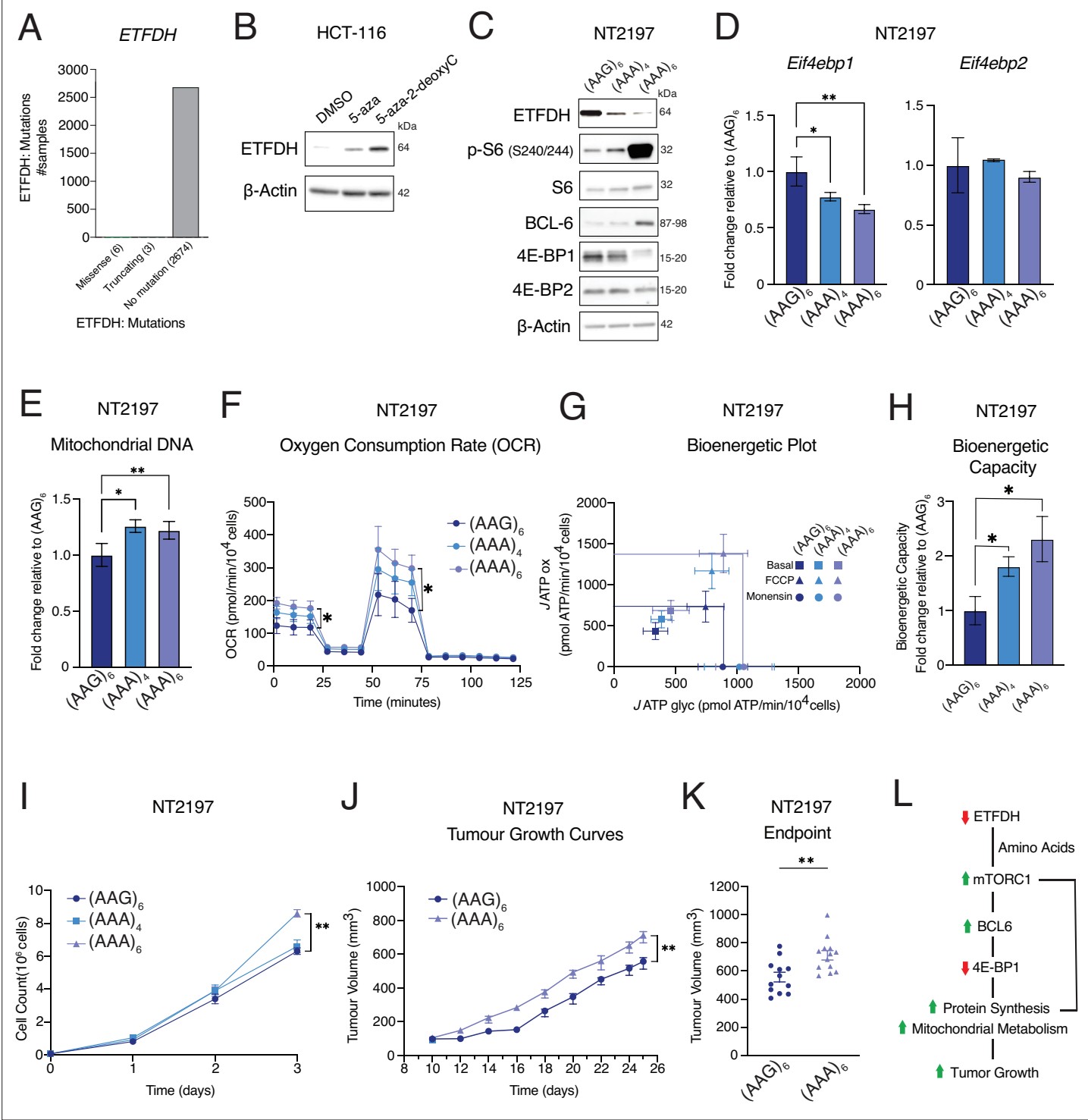

**Figure 6.** Reduced expression of ETFDH increases mitochondrial metabolism and tumor growth. (**A**) ETFDH mutations from 2683 samples across multiple cancer types. The number of missense and truncating mutations are shown relative to samples with no mutations. Data are acquired from the ICGC/TCGA Pan-Cancer Analysis of Whole Genomes Consortium (*Aaltonen et al., 2020*), accessed from the cBioPortal server (https://www.cbioportal.org). (**B**) Levels of indicated proteins in WT HCT-116 cells treated with 2.5 μM 5-azacytidine (5-aza), 10 μM 5-aza-2' -deoxycytidine (5-aza-2-deoxyC), or a vehicle (DMSO) for 72 hr were determined by western blotting. β-Actin was used as a loading control (representative blots of n=3). (**C**) Levels and the phosphorylation status of indicated proteins in ETFDH KO NT2197 cells expressing ETFDH harboring $(AAG)_6$, $(AAA)_4$, $(AAA)_6$ tracks were determined by western blotting. β-Actin was used as a loading control (representative blots of n=3). (**D**) *Eif4ebp1* and *Eif4ebp2* mRNA abundance in $(AAG)_6$, $(AAA)_4$, and $(AAA)_6$ NT2197 cells were determined by RT-qPCR. *Actb* was used as a housekeeping gene. Data are presented as fold change in *Eif4ebp1/Actb*

*Figure 6 continued on next page*

*Figure 6 continued*

and *Eif4ebp2/Actb* ratios relative to $(AAG)_6$ NT2197 cells (n=3), *$P$<0.05, **$P$<0.01, one-way ANOVA, Dunnett's post-hoc test. (**E**) Mitochondrial DNA content in $(AAG)_6$, $(AAA)_4$, and $(AAA)_6$ NT2197 cells were monitored by qPCR. Mitochondrial DNA (mtDNA) content was normalized to genomic DNA (gDNA) content. Data are presented as mean fold change relative to $(AAG)_6$ NT2197 cells -/+ SD (n=3), *$P$<0.05, **$P$<0.01, one-way ANOVA, Dunnett's post-hoc test. (**F**) Oxygen consumption of $(AAG)_6$, $(AAA)_4$, and $(AAA)_6$ NT2197 cells was determined using Seahorse bioanalyzer. Data are normalized to cell count and presented as means +/- SD (n=4), one-way ANOVA, Dunnett's post-hoc test. (**G–H**) Bioenergetic Plot for Basal, FCCP, and Monensin *J* ATP fluxes (**G**) and Bioenergetic Capacity (**H**) derived from $(AAG)_6$, $(AAA)_4$, and $(AAA)_6$ NT2197 cells. Data are presented as means +/- SD (n=4), one-way ANOVA, Dunnett's post-hoc test. (**I**) Proliferation of $(AAG)_6$, $(AAA)_4$, and $(AAA)_6$ NT2197 cells. ETFDH rescue measurements are the same as in *Figure 1— figure supplement 1F*. Data are presented as cell count means +/- SD (n=4), **$P$<0.01, one-way ANOVA, Dunnett's post-hoc test. (**J–K**) Tumor growth of $(AAG)_6$ and $(AAA)_6$ NT2197 cells following mammary fat-pad injection (**J**) and endpoint tumor volumes (**K**). Growth was assessed using calipers. Data are presented as means +/- SEM ($(AAG)_6$ n=6–12, $(AAA)_6$ n=11–13), **$P$<0.01, unpaired Student's *t* test. (**L**) Schematic representation of the model whereby reduction in ETFDH levels accelerates tumor growth. Reduction of ETFDH triggers accumulation of intracellular amino acids, thus increasing mTORC1 signaling. This is orchestrated with BCL-6-dependent reduction in 4E-BP1 levels to drive mitochondrial biogenesis and increase bioenergetic capacity of cancer cells, ultimately leading to more aggressive tumor growth.

The online version of this article includes the following source data and figure supplement(s) for figure 6:

**Source data 1.** PDF files containing original western blots for *Figure 6B and C*, indicating the relevant bands.

**Source data 2.** Original files for western blot analysis displayed in *Figure 6B and C*.

**Figure supplement 1.** Poly(A) Track design (**A–B**); Electron transfer flavoprotein dehydrogenase (ETFDH) catalytic activity is required for its tumor suppressive function (**C–F**).

**Figure supplement 1—source data 1.** PDF file containing original western blots for *Figure 6—figure supplement 1C*, indicating the relevant bands.

**Figure supplement 1—source data 2.** Original files for western blot analysis displayed in *Figure 6—figure supplement 1C*.

# Discussion

Notwithstanding that ETFDH mutations play a detrimental role in MADD metabolic disorders, we show that ETF insufficiency caused by reduced *ETFDH* expression plays a major role in rewiring cancer metabolism and signaling to stimulate neoplastic growth. Reduction of ETFDH levels appears to be a common feature of cancer cells and is likely a consequence of DNA hypermethylation (*Figure 6B*), an event that is frequently observed among tumor suppressor genes (*Jones and Baylin, 2002*). Indeed, ETFDH was found among the most under-expressed metabolic enzymes in a study that compiled cancer microarray datasets (*Nilsson et al., 2014*). Moreover, recent studies highlighted crosstalk between aberrant mTOR signaling and epigenetic perturbations (*Smith et al., 2019*; *Harachi et al., 2020*; *Chen et al., 2023*; *Kim et al., 2024*), which further reinforces the plausibility that epigenetic reprogramming in cancer cells may underpin the pro-neoplastic role of ETFDH downregulation and/ or explain resulting phenotypes.

Intriguingly, although ETF insufficiency limits the flexibility of cancer cells in using fuels (e.g. lipids and amino acids) for OXPHOS, it triggers signaling events that establish a feed-forward loop via mTORC1, leading to increased BCL-6 levels and reduced *EIF4EBP1* transcription (*Figure 6L*). ETF insufficiency thus orchestrates enhanced mTORC1 signaling with downregulation of 4E-BP1 protein levels to reprogram protein synthesis and ultimately increase mitochondrial biogenesis and functions, thereby providing a selective proliferative advantage to cancer cells (*Figure 6L*). To this end, ETF insufficiency allows cancer cells to trade their metabolic flexibility for increased bioenergetic capacity and a rewired signaling state that favors neoplastic growth. Importantly, while ETFDH plays an essential role in muscle, we did not observe a major effect of ETFDH disruption on proliferation, mTOR signaling or 4E-BP1 levels in non-transformed murine breast epithelial NMuMG cells. These observations suggest that ETF insufficiency causes remodeling of signaling and metabolic networks that are favourable to cancer cells.

Our findings are consistent with previously attributed tumor-suppressive effects of 4E-BP1 (*Petroulakis et al., 2009*; *Hsieh et al., 2010*; *Polunovsky et al., 2000*; *Lynch et al., 2004*; *Avdulov et al., 2004*; *Rousseau et al., 1996*). More recently, it has been demonstrated that under certain contexts, 4E-BPs may play a beneficial role for tumor survival. For instance, 4E-BPs may facilitate tumor growth by regulating global and selective mRNA translation under periods of cellular stress (*Musa et al., 2016*; *Jewer et al., 2020*), including suppression of fatty acid biosynthesis under glucose deprivation (*Teleman et al., 2005*; *Levy et al., 2024*). Collectively, these findings demonstrate that 4E-BPs are likely to play more complex, context-dependent roles during cancer progression than previously

anticipated. Moreover, the role of different 4E-BP proteoforms in cancer was, in general, thought to be redundant. Our findings, however, demonstrate that in the context of ETF insufficiency, selective decrease in 4E-BP1, but not 4E-BP2 levels, plays a role in driving neoplastic growth, thereby suggesting that 4E-BP1 and 4E-BP2 may play hitherto unappreciated non-overlapping roles in neoplasia.

Several metabolites can be sensed via signaling partners upstream of mTORC1, including leucine, arginine, methionine/SAM, and threonine (*Valenstein et al., 2025*). Branched-chain amino acids (leucine, isoleucine, and valine), which are among the highest upregulated metabolites in ETFDH-deficient cells (*Figure 3A*), serve as ETFDH substrates and have been described to display strong activation capabilities towards mTORC1 in the literature (*Appuhamy et al., 2012*; *Herningtyas et al., 2008*). Glutamine can also activate mTORC1 through the Arf family of GTPases (*Jewell et al., 2015*). Indeed, glutamine can supplement the non-essential amino acid (NEAA) pool through transamination (*Tan et al., 2017*) and amino acid uptake (*Chen et al., 2014*). Accordingly, the maintenance of NEAA that are non-ETFDH substrates may be supported by the global metabolic rewiring fueled by enhanced glutamine metabolism in ETFDH-deficient cells. Deciphering the mechanisms leading to accumulation of specific amino acids and their role in ETFDH-dependent mTORC1 modulation is warranted.

In conclusion, we show that ETF insufficiency caused by reduced ETFDH expression remodels signaling and mitochondrial metabolism in a manner that drives neoplastic growth (*Figure 6L*), while either being neutral (NMuMG) or deleterious (muscle cells) in non-malignant cells. These findings explain how disruption of a mitochondrial enzyme, rather than reducing cellular fitness, is common in cancer because it increases mitochondrial energy flux to drive neoplastic growth.

## Materials and methods
### Cell lines and cell culture
HCT-116 (RRID:CVCL_0291), NALM6 (RRID:CVCL_0092), and HEK293T (RRID:CVCL_0063) cells were obtained from American Type Culture Collection (ATCC) and verified using ATCC Cell line authentication service (STR profiling). NMuMG, NT2197, and MT4788 Parental and STAT1 KO cells were obtained by Dr. J. Ursini-Siegel's lab, while 4T1 (4T1-526) cells were obtained from Dr. P.M. Siegel's group. HEK293T and HCT-116 cells were cultured in DMEM supplemented with 10% fetal bovine serum (FBS), 1% penicillin/streptomycin and supplemented with 2 mM L-Glutamine to obtain a final concentration of 6 mM L-Glutamine. NALM6 and 4T1 cells were cultured in RPMI-1640 supplemented with 10% FBS, 1% penicillin/streptomycin, and 2 mM L-Glutamine. NMuMG and NT2197 cells were cultured in DMEM supplemented with 10% FBS, 1% penicillin/streptomycin, 2 mM L-Glutamine, 10 mg/mL insulin, 20 mM HEPES, pH 7.5. NT2197 culture media was supplemented with puromycin (2 µg/mL) as previously described (*Ursini-Siegel et al., 2008*). MT4788 were grown in DMEM, 2.5% FBS, mammary epithelial growth supplement (MEGS), 1% penicillin/streptomycin, and gentamycin as previously described (*Totten et al., 2021*). All cell lines were maintained at 37 °C and 5% CO2 in a humidified environment and tested periodically for mycoplasma according to the manufacturer's instructions.

### Animals
6–8 week-old SCID-BEIGE mice were purchased from Charles River Laboratories (Quebec, Canada, RRID:IMSR_CRL:250). Male SCID-BEIGE mice were used for intracaecal experiments, while female mice were used for mammary fat pad injections. All animal experiments were performed in accordance with the guidelines of the McGill University Animal Ethics Committee and the Canadian Council on Animal Care as approved by the facility animal care committee (protocol numbers MCGL-10212 and AUP#5129).

### Xenograft experiments
Mice were randomized prior to cell line injections. Luciferase-expressing HCT-116 cells (250,000 cells) were injected intracaecally in male (8–10 weeks old) SCID-BEIGE mice as previously described (*Tabariès et al., 2021*). Tumor growth was monitored and assessed by luminescence (IVIS spectrum, Perkin-Elmer). Animals were sacrificed after 50 days. Quantification of signal intensity was performed with Living Image software (PerkinElmer, RRID:SCR_014247). NT2197 mammary tumor cells (50,000 cells)

were injected in mammary fat pads of 6–8 week female SCID-BEIGE mice. Tumor growth was monitored using calipers. Measurements were shown as once tumor volume reached 100~200 mm$^3$ and animals were sacrificed before tumors reached 1 cm$^3$. Tumors were excised, placed in 10% formalin, and embedded for immunohistochemistry. No animals were excluded from experiments due to attrition or failure to complete the tumor volume measurements. Pilot experiments were carried out to estimate size effects pertinent to differences between tested groups/conditions. The number of animals/group that was used in the experiments was determined by calculating ~90% power to detect a ~40% difference in means between groups assuming a standard deviation of ~40% (alpha = 0.05).

## Cell proliferation assay

Cells were counted using a Countess automated cell counter (Invitrogen) and trypan blue exclusion was used for determining cell viability. Cells (HCT-116, NT2197, NMuMG, 4T1, NALM6) were seeded in 6-well dishes and counted after 3–5 days. HCT-116 cells were seeded on day 0 at a density of 100,000 cells for *Figure 1—figure supplement 1E*, *Figure 3—figure supplement 1C*, *Figure 3—figure supplement 1E–G*, *Figure 4—figure supplements 1D00* and 50,000 cells for *Figure 6—figure supplement 1D*. NT2197 cells were seeded on day 0 at a density of 60,000 cells in *Figure 6I*, *Figure 1—figure supplement 1F*. NMuMG cells were seeded on day 0 at a density of 60,000 cells in *Figure 1—figure supplement 1I*. 4T1 cells were seeded on day 0 at a density of 60,000 cells in *Figure 5I*, *Figure 1—figure supplement 1G*. NALM6 cells were seeded on day 0 at a density of 20,000 cells in *Figure 1—figure supplement 1H*. For glutamine deprivation experiments, HCT-116 cells were seeded in full culture for 24 hr. The next day, cells were grown in media in the presence or absence of glutamine for 48 hr and then counted. For Torin1/BiS-35x treatment experiments (*Figure 3—figure supplement 1C*; *Figure 3—figure supplement 1G*), HCT-116 cells were seeded overnight. The next day, media was replaced with treatment media containing Torin1 (50 nM, 100 nM, 250 nM, 500 nM) or BiS-35X (1 nM, 10 nM, 100 nM, 1000 nM) or DMSO for 72 hr and then counted. Counting was done as a single-blind study.

## Immunohistochemistry

Immunohistochemical staining of colon human paraffin-embedded tissue array (https://tissuearray.com/, CO992b) was performed as previously described (*Igelmann et al., 2021*). Briefly, the tissue slide was baked at 55 °C for 30 min followed by incubation in xylene (2×5 min), 100% ethanol (2×5 min), 95% ethanol (2×5 min), 75% ethanol (1×5 min), and ddH$_2$O (2×5 min). The slide was submerged in 10 mM sodium citrate buffer (pH 6.0), 0.05% Tween-20, and heat-induced epitope retrieval was carried out in a pressure cooker for 30 min. The slide was cooled for 20 min at room temperature, rinsed in ddH$_2$O for 5 min, and washed with 1xTBS +0.3% Triton X-100 for 10 min, then 1xTBS for 5 min at room temperature. The slide was incubated in 3% H$_2$O$_2$ for 10 min at room temperature and washed with TBST (1xTBS+0.1% Tween-20) for 5 min to inactivate peroxidases. A liquid blocker pen (PAP Pen, DAKO) was used to circle tissues on the slide. The TMA was blocked for 1 hr at room temperature in the presence of DAKO Protein Block-Serum Free (DAKO, Cat#X0909). The slide was incubated overnight at 4 °C with the ETFDH antibody (1:500, ProteinTech 11109–1-AP). The next day, the slide was washed in TBST (3×5 min), and secondary incubation was performed for 30 min at room temperature with secondary antibody (Antibody background reducing diluent and HRP rabbit substrate (Cell Signaling, Cat#7074)). The slide was washed in TBST (3×5 min) and DAB (Di-amine-benzidine) peroxidase substrate (Vector, Cat# SK-4100) was applied to the slide for 5 min. The reaction was stopped by washing in ddH$_2$O for 5 min. The slide was counterstained with filtered hematoxylin for 20 s and rinsed with ddH$_2$O for 5 min. The slide was placed in 75% ethanol (1 min), 95% ethanol (1 min), 100% ethanol (1 min), and xylene (2×5 min). The slide was mounted with a cytoseal at 60 °C and dried in a fume hood overnight. Quantitation of sections was performed using Imagescope software (Aperio; RRID:SCR_014247).

## Western blotting

Western blotting was performed as previously described (*Kim et al., 2024*). Briefly, cells were washed twice with PBS and lysed with RIPA buffer (50 mM Tris-HCl pH 7.5, 150 mM NaCl, 0.5% sodium deoxycholate, 1 mM PMSF, 1 mM DTT, 1% NP40, 0.1% SDS), supplemented with 1 X protease inhibitor (Roche) and 1 x PhoSTOP (Roche). Lysates were cleared at 4 °C (10 min, 13,000 rpm). Protein

concentrations were determined using BCA kit (Thermo Fisher). Lysates were mixed with 5 x Laemmli buffer (60 mM Tris-HCl (pH 6.8), 10% glycerol, 2% SDS, 5% 2-mercaptoethanol, 0.05% bromophenol blue) and boiled (95 °C for 5 min). Proteins were resolved by SDS-PAGE (Bio-Rad) and transferred via wet transfer apparatus (Bio-Rad) to nitrocellulose membranes (Amersham). Membranes were blocked in 5% BSA (dissolved in TBST (0.1% Tween 20 in 1xTBS)) and incubated in primary antibodies overnight at 4 °C. The next day, membranes were washed in TBST and incubated with HRP-conjugated secondary antibodies for 1 hr. Membranes were washed and applied with ECL (Bio-Rad) for 1 min. Exposure was carried out on an Azure c300 (Azure Biosystems). Images were quantified by ImageJ and analysed using GraphPad Prism 10; RRID:SCR_014247. The list of primary antibodies is described in *Supplementary file 1*.

For $\lambda$-phosphatase experiments, cells were lysed with RIPA and supplemented with 1 X protease inhibitor, without 1 x PhoSTOP. Protein samples were combined with NEBuffer for protein metallo-phosphatases (PMP), $MnCl_2$, and $\lambda$-protein phosphatase for 30 min at 30 °C according to the manufacturer's instructions (New England Biolabs).

## Cap-binding pull-down assay

The assay was carried out as previously described (*Dowling et al., 2010*). Briefly, NT2197 cells were seeded overnight in 150 mm plates. The next day, media was changed, and cells were washed with PBS and dissolved in lysis buffer containing 50 mM MOPS/KOH (7.4), 100 mM NaCl, 50 mM NaF, 2 mM EDTA, 2 mM EGTA, 1% NP40, 1% sodium deoxycholate, 7 mM β-mercaptoethanol, supplemented with 1 X protease inhibitor (Roche), and 1 x PhoSTOP (Roche). Lysates were cleared at 4 °C (10 min, 16,000 g). Protein concentrations in samples were elucidated by BCA (Thermo Fisher). Samples were equilibrated for 20 min rotating at 4 °C with $m^7$-GDP-agarose beads (Jena Bioscience), then washed in buffer containing 50 mM MOPS/KOH (7.4), 100 mM NaCl, 50 mM NaF, 0.5 mM EDTA, 0.5 mM EGTA, 7 mM β-mercaptoethanol, 0.5 mM PMSF, 1 mM $Na_3VO_4$, and 0.1 mM GTP, by centrifugation (500 g for 1 min). Bound proteins were eluted from beads via boiling in loading buffer. Western blotting was performed on eluted proteins and input samples to assess for eIF4F complex formation using 4E-BP1, eIF4G1, eIF4E, and β-actin antibodies.

## Flow cytometry

NT2197 cells were seeded in 6-well dishes for 24 hr. The next day, cells were stained with 50 nM MitoTracker Deep Red (Thermo Fisher) for 15 min at 37 °C in the dark. Cells were washed, counted, and 100,000 cells were dissolved in 500 µl HBSS (Gibco). Samples were analyzed with the LSR Fortessa cytometer (Becton Dickinson, Mountain View, CA). Fluorescence intensity was detected by excitation at 644 nm and acquisition on the 665 /-A channel for MitoTracker Deep Red. ROS measurements were performed as previously described (*Igelmann et al., 2021*). Briefly, cells were incubated with $H_2$DCFDA (Molecular Probes) for 30 min at 37 °C. Cells were trypsinized, washed, and resuspended in 500 µl HBSS. Samples were sorted on a BD FACS Canto II system. Fluorescence intensities were calculated by FlowJo (Tree Star, Inc; RRID:SCR_008520).

## Mitochondrial DNA quantitation

Quantitation of mitochondrial DNA was carried out as previously described (*Morita et al., 2013*). HCT-116 or NT2197 cells were seeded in 6-well dishes for 24 hr. Genomic and mitochondrial DNA were extracted using PureLink Genomic DNA Mini Kit (Thermo Fisher) and quantified by qPCR using SensiFAST SYBR Lo-ROX kit (Bioline).

## Polysome-profiling assay

Experiments were carried out as previously described (*Gandin et al., 2014*). Briefly, HCT-116 cells were seeded overnight in 150 mm plates. The next day, cells were washed in PBS containing 100 µg/ml cycloheximide and lysed in hypotonic buffer (5 mM Tris-HCl [pH 7.5], 2.5 mM $MgCl_2$, 1.5 mM KCl, 100 µg/ml cycloheximide, 2 mM DTT, 0.5% Triton X-100, and 0.5% sodium deoxycholate). Samples were loaded onto 10–50% wt/vol sucrose density gradients (20 mM HEPES-KOH pH 7.6, 100 mM KCl, 5 mM $MgCl_2$) and spun on a SW40Ti rotor (Beckman Coulter) at 4 °C (36,000 rpm for 2 hr). Samples were fractionated and recorded at OD 254 nm using an ISCO fractionator (Teledyne ISCO). RNA from fractions and input was extracted using TRIzol (Thermo Fisher) according to the manufacturer's

instructions. RNA from each fraction and input was isolated using TRIzol (Invitrogen) according to the manufacturer's instructions. RT-qPCR was performed on fractions and input RNA (refer to section 'RNA extraction and RT-qPCR' for more information). Primers are listed in *Supplementary file 2*.

## Puromycilation assay

HCT-116 cells were grown in 10 cm dishes for 24 hr. The next day, cells were treated with puromycin (10 µg/ml) for 20 min. Cells treated with DMSO were used as negative controls. Cells were lysed and protein samples were quantified. Western blotting was performed (refer to 'Western Blotting' section for more information) and membranes were incubated in anti-puromycin antibody (Millipore, Cat#MABE343). Quantification of bands was performed by ImageJ (RRID:SCR_003070).

## Stable isotope tracing

Metabolite detection by gas chromatography-mass spectrometry (GC-MS) and stable isotope tracing analysis (SITA) was performed as previously described (*Papadopoli et al., 2021*). Briefly, NT2197 cells were seeded in 6-well dishes. Prior to $^{13}$C-glutamine SITA experiments, culture media was replaced with one whose composition contains 6 mM unlabeled glutamine for 2 hr. Afterwards, media was replaced with media containing 6 mM labeled ([U-13C])-glutamine media (Cambridge Isotope Laboratories, MA, USA; CLM-1822; L-glutamine ([U-13C5]), 99%) for 5, 15, 30, and 60 min. For $^{13}$C-leucine SITA experiments, culture media was replaced with one containing unlabeled leucine (0.105 g/L) for 2 hr, then incubated in media containing labeled (0.105 g/L) ([U-13C])-leucine media (Cambridge Isotope Laboratories, MA, USA; CLM-2262-H-0.1; L-leucine ([U-13C6]), 99%) for 24 hr. Both steady state and tracing samples were washed in cold saline solution (9 g/L NaCl) and scraped off with 80% methanol. Samples were sonicated at 4 °C for 10 min (high setting, 30 s on/30 s off cycles) to rupture cells. Lysates were centrifuged at 4 °C (14,000 g, 10 min), and supernatants were collected with addition of an internal standard (750 ng myristic acid-D27). Samples were dried overnight at 4 °C by speed-vac (Labconco). Dried pellets are resuspended in methoxyamine hydrochloride (10 mg/ml), sonicated, and cleared for 10 min. Samples are heated to 70 °C for 30 min then applied with N-tert-butyldimethylsilyl-N-methyltrifluoroacetamide (MTBSTFA) at 70 °C for 1 hr. Derivatized samples are injected into an Agilent 5975 C GC/MS (Agilent Technologies, CA, USA) using methods described previously (*Papadopoli et al., 2021*). $^{13}$C-tracer samples were run in parallel with unlabeled samples. MassHunter software (Agilent) was used to carry out mass isotopomer distribution analyses, with metabolites normalized to myristic acid-D27 and cell number. Isotopic distributions were corrected for naturally occurring isotopes using an in-house algorithm (*McGuirk et al., 2013*).

## Steady-state analysis of nucleotides

Steady-state nucleotide abundances were determined using liquid chromatography-mass spectrometry (LC-MS/MS). Cells were washed with 150 mM ammonium formate solution (4 °C, pH 7.4), scraped, and extracted with 600 µl of methanol/acetonitrile solution (31.6% MeOH/36.3% ACN). Cell lysis and homogenization was conducted by bead-beating with four ceramic beads (2 mm) for 30 s at 50 Hz using a TissueLyser II (Qiagen). Dichloromethane was added to samples, which were subsequently centrifuged to separate extracts into aqueous and organic layers. Supernatants from aqueous layers were dried by vacuum centrifugation at –4 °C overnight (Labconco). Dried samples were re-suspended in 50 µl of water and centrifuged at 1 °C. 5 µl of sample was injected onto an Agilent 6470 Triple Quadrupole (QQQ)–LC–MS/MS (Agilent Technologies). Separation of metabolites was conducted using a 1290 Infinity ultra-performance quaternary pump liquid chromatography system, with a mass spectrometer equipped with a Jet Stream electrospray ionization source operating in negative mode (Agilent Technologies), with the following settings: source-gas temperature (150 °C), flow (13 L min$^{-1}$), nebulizer pressure (45 psi), and capillary voltage (2000 V). Chromatographic resolution of metabolites was conducted using a Zorbax Extend C18 column 1.8 µm, 2.1×150 mm$^2$ with guard column 1.8 µm, 2.1×5 mm$^2$ (Agilent Technologies). The gradient commenced at 100% mobile phase A (97% water, 3% methanol, 10 mM tributylamine, 15 mM acetic acid, 5 µM medronic acid) for 2.5 min, followed by a 5 min gradient to 20% mobile phase C (methanol, 10 mM tributylamine, 15 mM acetic acid, 5 µM medronic acid), a 5.5 min gradient to 45% C and a 7 min gradient to 99% C at a flow rate of 0.25 mL min$^{-1}$. This was followed by a 4 min hold time at 100% mobile phase C. The column was restored by washing with 99% mobile phase D (90% ACN) for 3 min at 0.25 mL min$^{-1}$, followed by

an increase of the flow rate to 0.8 mL min⁻¹ over 0.5 min and a 3.85 min hold, after which the flow rate was decreased to 0.6 mL min⁻¹ over 0.15 min. The column was subsequently re-equilibrated at 100% A over 0.75 min, during which the flow rate was decreased to 0.4 mL min⁻¹ and held for 7.65 min. The flow was brought back to forward flow at 0.25 mL min⁻¹ 1 min before the next injection. The column temperature was maintained at 35 °C. Data were analyzed by using MassHunter Quantitative Data Analysis B.10.00 (Agilent Technologies). Data presented are peak area normalized by cell number. Authentic metabolite standards (Sigma Aldrich) were used to obtain reaction monitoring parameters (qualifier/quantifier ions and retention times). All LC/MS grade solvents and salts are purchased from Fisher.

## RNA extraction and RT-qPCR

HCT-116 and NT2197 cells were plated in 6-well dishes and extracted using TRIzol (Ambion) or Aurum Total RNA Mini Kit (Bio-Rad) following the manufacturer's instructions. cDNA was synthesized from purified RNA using SensiFAST cDNA Synthesis Kit (Bioline), as per the manufacturer's protocol. RT-qPCR was performed using SensiFAST SYBR Lo-ROX kit (Bioline). Primer sequences are found in *Supplementary file 2*.

## Chromatin immunoprecipitation (ChIP)

Experiments were performed as previously described (*Kim et al., 2024*). Briefly, HCT-116 cells were grown in 150 mm dishes and fixed in 4% formaldehyde for 10 min, followed by centrifugation. Pellets were resuspended in ChIP buffer (50 nM Tris (pH 8), 100 mM NaCl, 5 mM EDTA, 1 X PMSF, 2 mM NaF, 0.25% NP-40, 0.25% Triton X-100, 0.25% Sodium Deoxycholate, 0.005% SDS) supplemented with 1 x cOmplete protease inhibitor (Roche). Samples are sonicated using Sonic Dismembrator Model 500 (Thermo Fisher) at 5 cycles at 20% power, 5 cycles at 25% power, five cycles at 30% power; each cycle is 10 s. Lysates are spun at 4 °C (14,000 rpm, 10 min), and protein concentration was measured from supernatants using the BCA kit (Thermo Fisher). Samples were loaded onto Protein G Plus-Agarose Suspension Beads (Millipore) and pre-cleared at 4 °C for 3 hours. Input samples were collected, and immunoprecipitation was performed with anti-BCL6 (Cell Signaling, #5650) or anti-IgG (Cell Signaling, #2729) on the remainder of the sample overnight at 4 °C. The following day, samples were washed in three wash buffers (20 mM Tris (pH 8), 2 mM EDTA, 0.10% SDS, 1% Triton X-100 with 150/200/599 mM for Wash1/2/3 buffers, respectively), then a wash with LiCl buffer (10 mM Tris (pH 8), 1 mM EDTA, 0.25 M LiCl, 1% NP-40, 1% Sodium Deoxycholate), and two washes with TE buffer (10 mM Tris (pH 8), 1 mM EDTA). Samples were eluted in elution buffer (0.1 M NaHCO3, 1% SDS) and de-crosslinked at 65 °C overnight. The following day, proteinase K was applied to samples and heated at 42 °C for 1 hr. DNA was purified and collected using a DNA collection column (BioBasic). ChIP-qPCR was performed for *EIF4EBP1* and *EIF4EBP2* using sequences found in *Supplementary file 3*.

## Respirometry assay

Oxygen consumption (OCR) and extracellular acidification rate (ECAR) were measured using Seahorse XFe24 and XFe96 analyzers (Agilent) with the Mito Stress Test (Agilent) and Palmitate Oxidation Assay (Agilent), according to the manufacturer's instructions. Briefly, HCT-116 and NT2197 cells were seeded in Seahorse culture plates (XFe24: 50,000 cells or XFe96: 20,000 cells) and incubated at 37 °C. Cells were washed twice and incubated in assay media. For mito stress test experiments, assay media is composed of 10 mM glucose, 2 mM glutamine, and 1 mM sodium pyruvate. Injections include oligomycin (1 µM), FCCP (1 µM), rotenone/antimycin A (1 µM), and monensin (20 µM). ATP production from OXPHOS (ATPox) or glycolysis (ATPglyc) were quantified using algorithms presented previously (*Mookerjee et al., 2017*). For the palmitate oxidation assay, cells were seeded in substrate-limited media (DMEM, 1% FBS, 0.5 mM glucose, 1 mM glutamine, 0.5 mM carnitine). The next day, cells are washed with FAO Buffer (includes 0.5 mM glucose, 0.5 mM carnitine, and 5 mM HEPES). Immediately prior to the run, palmitate-BSA or BSA control is added to the cell media. Basal respiration was recorded and values were normalized to cell counts.

## Soft agar/colony formation assay

Experiments were conducted as previously shown (*Borowicz et al., 2014*). Briefly, 6-well dishes are filled with a bottom layer solution (1:1; 1% noble agar and 2 X culture media (2 X DMEM, 20% FBS,

2% Pen/Strep/Glutamine)) per well. Dishes are incubated at room temperature for 30 min to allow the solution to solidify. 5000–10,000 HCT-116 and NT2197 cells are mixed 1:1 with 0.6% noble agar and 2 X culture media. 1.5 ml of this mixture is transferred on top of the solidified agar wells. Dishes are incubated at room temperature to allow the top layer to solidify. Cells are grown at 37 °C until colony formation is present. Dishes were stained with nitroblue tetrazolium (Sigma), and colonies were counted. Counting was done as a single-blind study.

## Protein stability assay

HCT-116 cells were seeded in 10 cm dishes. The next day, cells were treated with cycloheximide (CHX) (50 µg/ml) or DMSO for 2 hr. Furthermore, cells were treated with the proteasomal inhibitor, MG-132 (10 µM) or DMSO for 2 hr. Final time point for all conditions was 4 hr. Protein was extracted, quantified, and western blotting was performed for ETFDH, p21, 4E-BP1, or β-actin (refer to 'Western Blotting' section for more information).

## RNA stability assay

The assay was performed as previously described (*Topisirovic et al., 2009*). Briefly, HCT-116 cells were seeded in 6-well dishes. The next day, cells were treated with actinomycin D (ActD) for 2, 4, 8, and 24 hr. RNA was extracted, cDNA was synthesized, and RT-qPCR was performed (refer to 'RNA extraction and RT-qPCR' for more information).

## Generation of ETFDH knockout cell lines

To deplete endogenous ETFDH expression in HCT-116 and NALM6 cells, two guide RNAs (gRNAs) targeting exon 6 and exon 9 of human ETFDH (against ETFDH sequences AGGTTGGCCGAATGCT AGGATGG and GATGTAGGGATACAAAAGGATGG) (refer to *Supplementary file 4* for gRNA sequences) were designed using CHOPCHOP (https://chopchop.cbu.uib.no/) (*Labun et al., 2019*) and purchased with the appropriate overhangs to be cloned into LentiCRISPRv2(GFP) (RRID:Addgene_82416). Cloning was carried out as previously described (*Sanjana et al., 2014*; *Shalem et al., 2014*), with the following modifications: BsmBI-v2 (New England Biolabs) was used to digest the lentiCRISPRv2 plasmid; plasmid DNA was recovered using Zymoclean Gel DNA Recovery Kit (Zymo Research), and ligation was performed overnight at 4 °C. Transformation of the ligated plasmid was carried out at 42 °C in chemically competent Stbl3 *E. coli* (Thermo Fisher Scientific, C737303). Bacteria were plated on Lysogeny Broth (LB)-agar plates containing ampicillin (1 µg/mL) (BioBasic). Extraction of plasmid DNA was conducted using the QIAprep Spin Miniprep Kit (Qiagen), and samples were sequenced using primer for human U6 promoter (refer to *Supplementary file 5* for sequence). Nucleofection was used to deliver the validated lentiCRISPRv2 plasmids into HCT-116 and NALM6 cells. Transfections of the plasmids were carried out in Nucleofector cuvettes, according to the optimized protocols for NALM6 and HCT-116 cells (Lonza Bioscience), with $1\times10^6$ cells used and 1 X PBS used instead of the commercial 4-D Nucleofector solution. Post-nucleofection, cells were allowed to grow for 72 hr at 37 °C, 5% $CO_2$, prior to cell sorting (BD FACSAria Fusion Flow Cytometer (BD Biosciences)). Cells expressing GFP were selected and grown in 96-well plates containing conditioned media (45% filtered culture media, 45% fresh media, and 10% FBS). After two weeks, sorted clones were transferred into 24-well plates and later into 6-well plates to be collected and validated. Depletion of endogenous ETFDH protein levels in both HCT-116 and NALM6 cells was initially confirmed by immunoblotting. Candidate clones with loss of ETFDH expression were kept for further validation. Moreover, the genomic DNA loci of candidate clones were sequenced and analyzed for insertion–deletion mutations (INDELs). Two sets of PCR primers were designed in regions flanking the CRISPR target sites (refer to *Supplementary file 5* for validation PCR sequences). Extraction of genomic DNA for each clone was carried out using the PureLink Genomic DNA Mini Kit (Fisher Scientific). Two rounds of polymerase chain reaction (PCR) on the extracted genomic DNA (35 PCR cycles with Set 1 followed by 45 PCR cycles with Set 2) were carried out and gel electrophoresis was utilized to separate the PCR products. The smaller PCR product for each clone was excised and purified using the Zymoclean Gel DNA Recovery Kit (Zymo Research). The products were sequenced, and chromatograms were analyzed for the presence of mutations causing a premature stop codon or a frameshift. Insertion–deletion mutations (INDELs) in *ETFDH* were verified by PCR followed by sequencing, while the absence of ETFDH protein was confirmed by western blotting for HCT-116 cells and NALM6 cells.

To deplete ETFDH expression in NMuMG, NT2197, and 4T1 cells, two guides (gRNAs) targeting exon 2 of mouse Etfdh (against mEtfdh sequences 5'-ATTTTTATGCAGCGTATCACTGG-3' and 5'-GAACATCTTGGAGCACACAGAGG-3') were designed using CHOPCHOP and cloned in LentiCRISPRv2(GFP) (Addgene #82416) and sequenced as shown above. For lentiviral production, $2.5 \times 10^5$ HEK293T cells were seeded overnight in a 6 cm dish (Sarstedt). The next day, cells were co-transfected with 4 µg of respective lentiCRISPRV2 GFP plasmid containing either one of the gRNA inserts or empty-lentiCRISPRV2 GFP plasmid, 2.66 µg of psPAX2 packaging plasmid (RRID:Addgene_35002), and 1.66 µg of pMD2.G plasmid (RRID:Addgene_12259) using the jetPRIME transfection reagent as described by the manufacturer's protocol (Polyplus transfection). Growth media was changed after 24 hr. After 48 hr, viral supernatant was filtered (0.45 µm filter; (Frogga Bio)), mixed 1:1 with fresh culture media, and applied to target cells grown in 6-well plates along with 8 µg/mL polybrene (Sigma-Aldrich). Cells were re-transduced the following two days with viral supernatant. Cells were allowed to recover for 48 hr post-transduction. Cells were sorted into 96-well plates (Sarstedt) and grown to generate stable cell lines, as shown above. Depletion of mouse Etfdh protein levels was initially confirmed via immunoblotting for NT2197, 4T1, and NMuMG. The presence of INDELs leading to a premature stop codon or frameshift was determined by sequencing as shown above. Two sets of PCR primers were designed in the regions flanking the CRISPR target sites (refer to *Supplementary file 5*).

## Generation of ETFDH rescue cell lines

To rescue human ETFDH expression in HCT-116 cells, ETFDH(NM_004453) Human Tagged ORF was cloned into pLenti-ETFDH-C-Myc-DDK-P2A-Puro plasmid according to the manufacturer's instructions (Origene) to generate the resulting plasmid, pLenti-ETFDH-C-Myc-DDK-P2A-Puro. To re-express mutant ETFDH (Y304A, G306E), WT ETFDH, or ORF-Stuffer lentiviral plasmids conferring hygromycin B resistance were designed using VectorBuilder (VB250613-1532mdm, VB250613-1530fzr, and VB900172-2887sfj, respectively). Plasmids were transfected with psPAX2, pMD2.G into HEK293T cells with JetPrime reagent (Polyplus) according to the manufacturer's instructions. Viral supernatants were filtered (0.45 µM) and applied to HCT-116 ETFDH KO cells with polybrene (6 µg/mL). After 6 hr, cells were re-infected overnight. After 2 days, selection with puromycin (4 µg/ml) was performed for 72 hr. Uninfected cells were selected with puromycin to serve as negative controls. The reintroduction of ETFDH expression or control plasmid in the knockout cells was confirmed via western blotting.

To rescue ETFDH expression in NT2197 ETFDH KO cells, a lentiviral plasmid conferring hygromycin B resistance and expressing mEtfdh with PAM site mutations or expressing an ORF-Stuffer was created using VectorBuilder (VB221206-1257xet and VB900123-2599cba, respectively). Lentivirus production and transduction were carried out as previously mentioned for three consecutive days. After 48 hr of recovery, cells were selected with 1 mg/mL hygromycin B for 3 days. Cells were confirmed by Western blotting.

## Generation of Poly(A) track ETFDH variants

Mutants were designed by inserting poly(A) tracks of multiple adenosines: control track (CTRL) [six consecutive lysine AAG codons $(AAG)_6$], 12 adenosines [12 A, equivalent to four consecutive AAA (Lys) codons $(AAA)_4$]; 18 adenosines [18 A, equivalent to six consecutive AAA (Lys) codons $(AAA)_6$] within the rescue plasmid pLV[Exp]-Hygro-EF1A>mEtfdh. The polyA track insertion site was determined using AlphaFold (version 2) (*Jumper et al., 2021*; *Varadi et al., 2022*) to minimize unintended effects caused by the poly-lysine stretches on the possible mitochondrial import sequence. The polyA tracks were inserted following residue 38, in an unstructured loop of the Etfdh protein. Site-directed mutagenesis was performed using the QuikChange II site-directed mutagenesis kit, according to the manufacturer's instructions (Agilent). Briefly, cycling parameters are followed as: Cycle 1: 95 °C for 1 min; Cycle 2: 95 °C for 30 s, 60 °C for 30 s, 68 °C for 10 min; Cycle 3: 68 °C for 7 min. The resulting plasmids were generated: pLV[Exp]-Hygro-EF1A>mEtfdh-$AAG_6$, pLV[Exp]-Hygro-EF1A>mEtfdh-$AAA_4$, pLV[Exp]-Hygro-EF1A>mEtfdh-$AAA_6$. To generate the poly(A) track Etfdh mutant lines in NT2197 cells (termed $AAG_6$, $AAA_4$, $AAA_6$), lentivirus production and transduction into NT2197 ETFDH KO cells was carried out as previously mentioned for three consecutive days. After 48 hr of recovery, cells were selected with 1 mg/mL hygromycin B for 3–4 days. ETFDH expression was confirmed by western blotting.

## Generation of cells expressing *EIF4EBP1*

pBABE-puro-EV and pBABE-puro-4E-BP1 plasmids were used to overexpress 4E-BP1 in HCT-116 ETFDH KO cells. Retrovirus production and transduction was carried out similarly to lentivirus production, except 2.66 µg of pUMVC packaging plasmid (RRID:Addgene_8449) was utilized instead of psPAX2 (RRID:Addgene_12260). Following three consecutive days of viral transduction, cells were allowed to recover for 48 hr prior to selection with 2 µg/mL puromycin (Sigma-Aldrich) for 3 days. To overexpress 4e-bp1 in NT2197 ETFDH KO cells, plasmids containing m4e-bp1 or ORF-Stuffer were generated by VectorBuilder (VB221206-1246sbx and VB900123-2599cba). Lentivirus production and transduction was carried out as previously mentioned for three consecutive days prior to a 48 hr recovery period. After this period, selection with hygromycin B (1 mg/mL) was conducted for 5–7 days. Validation was observed by western blotting.

## Generation of knockdown models by shRNA/siRNA

Knockdown using shRNA was performed as previously described (*Hulea et al., 2018*). For BCL6 knockdown experiments, the following shRNA vectors were used: pLKO.1 Non-Target shRNA Control (Sigma, SHC002), BCL6 shRNA #1 (Sigma, TRCN0000084654), or BCL6 shRNA #2 (TRCN0000084655). For RAPTOR/RICTOR knockdown experiments, the following shRNA vectors were used: scramble shRNA (Addgene, Plasmid #1864), RAPTOR_1 shRNA (Addgene, Plasmid #1857), and RICTOR_1 shRNA (Addgene, Plasmid #1853). Briefly, HEK293T cells were transfected with psPAX2, pMD2.G, or target pLKO.1 lentiviral shRNA vectors with JetPrime reagent (Polyplus) according to the manufacturer's instructions. Viral supernatants were filtered (0.45 µM) and transferred to target cells with polybrene (6 µg/mL). After 6 hr, cells were re-infected overnight. After 2 days, selection with puromycin (4 µg/ml) was performed for 72 hr. Uninfected cells were selected with puromycin to serve as negative controls. Cells were seeded and processed for protein extraction or cell proliferation. Validation was confirmed by western blotting for *BCL6* knockdown and *RAPTOR* or *RICTOR* knockdown.

Knockdown using siRNA was performed as previously described (*Papadopoli et al., 2021*). Briefly, target cells were transfected for 24 hr with 50 nM Silencer Negative Control No.1 (Thermo Fisher, AM4611) or BCL6 siRNA (Thermo Fisher, Cat#4390771) using JetPrime reagent (Polyplus) according to the manufacturer's instructions. Cells were re-transfected the next day for 24 hr, then processed for protein extraction. Validation was confirmed by western blotting.

## Expression profiles from TCGA/GTEx databases

Expression of *ETFDH* and *EIF4EBP1* mRNA across both normal and tumor samples were obtained from The Cancer Genome Atlas (TCGA) and GTEx repositories. The expression data were normalized using the TMM (trimmed mean of M values) method and presented as log2 counts per million. Correlations between the genes were assessed using Pearson's correlation coefficient.

## Quantification and statistical analysis

Statistical analysis was performed using GraphPad Prism 10 (RRID:SCR_002798). Data are presented as mean +/- SD of independent experiments, unless stated otherwise. Technical replicates were averaged from two to three independent experiments. Details on data quantification, presentation, and statistical analysis are included in figure legends.

## Material and correspondence

Further information and requests for reagents may be directed upon reasonable request by the lead contact, Ivan Topisirovic (ivan.topisirovic@mcgill.ca).

## Acknowledgements

We are thankful to the members of all involved laboratories and Stephane Richard for helpful discussions. We thank Luc Choinière and Mariana Russo for assistance with metabolomic experiments, Christian Young for assistance with flow cytometry, and Christopher Rudd for assistance with respirometry. This research was funded by the Terry Fox Foundation (TFF) Oncometabolism Team Grant (TFF-242122) to IT, PS, and MP, and Canadian Institutes of Health Research (CIHR) (PJT-183843, PJT-479494) to IT. Research in OL's lab is supported by grants from the Swedish Research Council (2020-01665), Swedish

Cancer Society (22 2186), the Cancer Research Funds of Radiumhemmet (231263) and the Wallenberg Academy Fellow Program. L-MP acknowledges the funds from Queen's University. DP is supported by the CIHR Postdoctoral Fellowship (MFE-171312), Cancer Research Society (CRS) The Next Generation of Scientists Award (NGS), and LDI Desjardins Fellowship Program. PJ and HK are supported by Fonds de Recherche du Québec – Santé (FRQS) Fellowships. IT is supported by the Canada Research Chair in Regulation of mRNA Translation and Metabolism. Metabolic analysis was performed at the Rosalind and Morris Goodman Cancer Research Centre's Metabolomics Core Facility, which is supported by the Canada Foundation for Innovation, the Dr. John R and Clara M Fraser Memorial Trust, the Terry Fox Foundation (TFF Oncometabolism Team Grant; TFF-242122), and McGill University.

## Additional information

### Competing interests

Lynne-Marie Postovit, Ivan Topisirovic: Reviewing editor, eLife. The other authors declare that no competing interests exist.

### Funding

| Funder | Grant reference number | Author |
| --- | --- | --- |
| Canadian Institutes of Health Research | PJT-183843 | Ivan Topisirovic |
| Canadian Institutes of Health Research | PJT-479494 | Ivan Topisirovic |
| Terry Fox Research Institute | TFF-242122 | Peter M Siegel Michael Pollak Ivan Topisirovic |
| Cancer Research Society | NGS | David Papadopoli |
| Canada Research Chairs | Tier 1 | Ivan Topisirovic |
| Canadian Institutes of Health Research | MFE-171312 | David Papadopoli |
| Cancerfonden | 22 2186 | Ola Larsson |
| Swedish Research Council | 2020-01665 | Ola Larsson |
| the Cancer Research Funds Radiumhemmet | 231263 | Ola Larsson |
| Queen's University | | Lynne-Marie Postovit |
| Fonds de Recherche du Québec – Santé (FRQS) Fellowships | | HaEun Kim Predrag Jovanovic |

The funders had no role in study design, data collection and interpretation, or the decision to submit the work for publication.

### Author contributions

David Papadopoli, Conceptualization, Data curation, Formal analysis, Funding acquisition, Validation, Investigation, Visualization, Methodology, Writing – original draft, Writing – review and editing; Ranveer Palia, HaEun Kim, Investigation, Methodology, Writing – review and editing; Predrag Jovanovic, Sebastian Igelmann, Formal analysis, Investigation, Methodology, Writing – review and editing; Sébastien Tabariès, Emma Ciccolini, Formal analysis, Investigation, Writing – review and editing; Valerie Sabourin, Shannon McLaughlan, Lesley Zhan, Nabila Chekkal, Jibin Zeng, Julia Vassalakis, Farzaneh Afzali, Slim Mzoughi, Investigation, Writing – review and editing; Krzysztof J Szkop, Formal analysis, Writing – review and editing; Thierry Bertomeu, Formal analysis, Methodology, Writing – review and editing; Ernesto Guccione, Mike Tyers, Daina Avizonis, Lynne-Marie Postovit, Sergej Djuranovic, Resources, Writing – review and editing; Ola Larsson, Peter M Siegel, Resources, Funding acquisition, Writing – review and

editing; Josie Ursini-Siegel, Supervision, Writing – review and editing; Michael Pollak, Conceptualization, Resources, Supervision, Funding acquisition, Writing – original draft, Project administration, Writing – review and editing; Ivan Topisirovic, Conceptualization, Resources, Supervision, Funding acquisition, Methodology, Writing – original draft, Project administration, Writing – review and editing

## Author ORCIDs
David Papadopoli ⓘ https://orcid.org/0000-0002-1248-1839
Ranveer Palia ⓘ https://orcid.org/0000-0003-4265-243X
Thierry Bertomeu ⓘ https://orcid.org/0000-0002-5313-7057
Julia Vassalakis ⓘ https://orcid.org/0009-0000-1261-3638
Ernesto Guccione ⓘ https://orcid.org/0000-0001-7764-5307
Lynne-Marie Postovit ⓘ https://orcid.org/0000-0002-8088-4197
Sergej Djuranovic ⓘ https://orcid.org/0000-0002-9417-0822
Peter M Siegel ⓘ https://orcid.org/0000-0002-5568-6586
Michael Pollak ⓘ https://orcid.org/0000-0003-3047-0604
Ivan Topisirovic ⓘ https://orcid.org/0000-0002-5510-9762

## Ethics
All animal experiments were performed in accordance with the guidelines of the McGill University Animal Ethics Committee and the Canadian Council on Animal Care as approved by the facility animal care committee. Protocol number MCGL-10212 and AUP-5129.

Reviewer #1 (Public review): https://doi.org/10.7554/eLife.106587.3.sa1
Reviewer #2 (Public review): https://doi.org/10.7554/eLife.106587.3.sa2
Author response https://doi.org/10.7554/eLife.106587.3.sa3

# Additional files

## Supplementary files
Supplementary file 1. List of Antibodies.

Supplementary file 2. Primer sequences used for RT-qPCR.

Supplementary file 3. Primer sequences used for ChIP-qPCR.

Supplementary file 4. gRNA sequences used for generation of electron transfer flavoprotein dehydrogenase (ETFDH) knockout (KO) cell lines.

Supplementary file 5. Primer sequences used for sequencing.

MDAR checklist

## Data availability
Plasmids generated in this study are deposited on Addgene. The original data are available at Mendeley Data (doi: 10.17632/p4wxkr3vdm.1). Metabolomics data are available at MetaboLights (MTBLS11048).

The following datasets were generated:

| Author(s) | Year | Dataset title | Dataset URL | Database and Identifier |
|---|---|---|---|---|
| Papadopoli D | 2026 | Mitochondrial ETF insufficiency drives neoplastic growth by selectively optimizing cancer bioenergetics | https://doi.org/10.17632/p4wxkr3vdm.1 | Mendeley Data, 10.17632/p4wxkr3vdm.1 |
| Papadopoli D | 2026 | Mitochondrial ETF insufficiency drives neoplastic growth by selectively optimizing cancer bioenergetics | https://www.ebi.ac.uk/metabolights/MTBLS11048 | Metabolights, MTBLS11048 |

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
