## [Editor Report · eLife Assessment]

The authors present an **important** set of data implicating ETFDH as an epigenetically suppressed gene in cancer with tumor suppressive functions. The evidence is **convincing**, with the authors demonstrating that suppression of ETFDH activity results in accumulation of amino acids that impact metabolism via hyperactive mTORC1.

---

## [Referee Report · Reviewer #1 (Public review)]

In their manuscript, Papadopoli et al explore the role of ETFDH in transformation. They note that ETFDH protein levels are decreased in cancer, and that deletion of ETFDH in cancer cell lines results in increased tumorigenesis, elevated OXPHOS and glycolysis, and a reduction in lipid and amino acid oxidation. The authors attribute these effects to increased amino acid levels stimulating mTORC1 signaling and driving alterations in BCL6 and EIF4EBP1. They conclude that ETFDH1 is epigenetically silenced in a proportion of neoplasms, suggesting a tumor-suppressive function. Overall, the authors logically present clear data and perform appropriate experiments to support their hypotheses.

---

## [Referee Report · Reviewer #2 (Public review)]

Summary:

The altered metabolism of tumors enables their growth and survival. Classically, tumor metabolism often involves increased activity of a given pathway in intermediary metabolism to provide energy or substrates needed for growth. Papadopoli et al. investigate the converse - the role of mitochondrial electron transfer flavoprotein dehydrogenase (ETFDH) in cancer metabolism and growth. The authors present compelling evidence that ETFDH insufficiency, which is detrimental in non-malignant tissues, paradoxically enhances bioenergetic capacity and accelerates neoplastic growth in cancer cells in spite of the decreased metabolic fuel flexibility that this affords tumor cells. This is achieved through the retrograde activation of the mTORC1/BCL-6/4E-BP1 axis, leading to metabolic and signaling reprogramming that favors tumor progression.

Strengths:

This review focuses primarily on the cancer metabolism aspects of the manuscript.

The study provides robust evidence linking ETFDH insufficiency to enhanced cancer cell bioenergetics and tumor growth.

The use of multiple cancer cell lines and *in vivo* models strengthens the generalizability of the findings.

The mechanistic insights into the mTORC1/BCL-6/4E-BP1 axis and its role in metabolic reprogramming are of general interest within and outside the immediate field of tumor metabolism.

Conclusion:

This manuscript provides significant insights into the role of ETFDH insufficiency in cancer metabolism and growth. The findings highlight the potential of targeting the mTORC1/BCL-6/4E-BP1 axis in ETFDH-deficient cancers. The compelling data support the conclusions presented in the manuscript, which will be valuable to the cancer metabolism community.

[Editors' note: The authors have addressed each of the two weaknesses previously listed in the public review, providing new experimental data on nucleotides and showing that the catalytic activity is required via the suggested addback experiment.]

---

## [Author Response]

The following is the authors’ response to the original reviews.

**Public Reviews:**

**Reviewer #1 (Public review):**
Authors state, "we identified ETF dehydrogenase (ETFDH) as one of the most dispensable metabolic genes in neoplasia." Surely there are thousands of genes that are dispensable for neoplasia. Perhaps the authors can revise this sentence and similar sentiments in the text.

We agree with the reviewer and have corrected the text accordingly. Specifically, we rephrased the sentence: “Surprisingly, we observed that in contrast to muscle, ETFDH is one of the most non-essential metabolic genes in cancer cells.” to “Surprisingly, we observed that in contrast to muscle, ETFDH is a non-essential gene in acute lymphoblastic leukemia NALM-6 cells”

Authors state, "These findings show that ETFDH loss elevates glutamine utilization in the CAC to support mitochondrial metabolism." While elevated glutamine to CAC flux is consistent with the statement that increased glutamine, the authors have not measured the effect of restoring glutamine utilization to baseline on mitochondrial metabolism. Thus, the causality implied by the authors can only be inferred based on the data presented. Indeed, the increased glutamine consumption may be linked to the increase in ROS, as glutamate efflux via system xCT is a major determinant of glutamine catabolism in vitro.

Indeed. We changed the statement "These findings show that ETFDH loss elevates glutamine utilization in the CAC to support mitochondrial metabolism." to "Collectively, these data demonstrate that ETF insufficiency in cancer cells remodels mitochondrial metabolism and increases the glutamine consumption and anaplerosis."

Authors state that the mechanism described is an example of "retrograde signaling". However, the mechanism seems to be related to a reduction in BCAA catabolism, suggesting that the observed effects may be a consequence of altered metabolic flux rather than a direct signaling pathway. The data presented do not delineate whether the observed effects stem from disrupted mitochondrial communication or from shifts in nutrient availability and metabolic regulation.

Notwithstanding that the term “retrograde” was used to refer to signaling from mitochondria to mTORC1, rather than from mTORC1 to mitochondria [1], we have removed the term “retrograde signaling” throughout the manuscript.

The authors should discuss which amino acids that are ETFDH substrates might affect mTORC1 activity or consider whether other ETFDH substrates might also affect mTORC1 in their discussion. Along these lines, the authors might consider discussing why amino acids that are not ETFDH substrates are increased upon ETFDH loss.

Based on the literature, we expect that branched chain amino acids that are ETFDH substrates (e.g., leucine) are likely to play a major role in activating mTORC1 upon ETFDH abrogation. As expected, the aforementioned amino acids are among those that are the most highly upregulated in ETFDH deficient cells (Fig 3A). We have, however, never formally tested the role of branched chain amino acid in activating mTORC1 in the context of ETFDH disruption. The increase in amino acids that are not metabolized via ETFDH, is likely to stem from global metabolic rewiring of ETFDH-deficient cells and observed alterations in amino acid uptake (e.g., glutamine; Fig 2F). We discuss this in the revised version of the paper as follows:

“Several metabolites can be sensed via signaling partners upstream of mTORC1, including leucine, arginine, methionine/SAM, and threonine [2]. Branched-chain amino acids (leucine, isoleucine, and valine), which are among the highest upregulated metabolites in ETFDH deficient cells (Fig 3A) serve as ETFDH substrates, and have been described to display strong activation capabilities towards mTORC1 in the literature [3,4]. Glutamine can also activate mTORC1 through Arf family of GTPases [5]. Indeed, glutamine can supplement the non-essential amino acid (NEAA) pool through transamination [6] and amino acid uptake [7]. Accordingly, the maintenance of NEAA that are non-ETFDH substrates may be supported by the global metabolic rewiring fueled by enhanced glutamine metabolism in ETFDH-deficient cells. Deciphering the mechanisms leading to accumulation of specific amino acids and their role in ETFDH-dependent mTORC1 modulation is warranted.”

**Reviewer #2 (Public review):**
The authors would strengthen the paper considerably by adding back catalytically inactive ETFDH to show that the activity of this enzyme is responsible for the increased growth phenotypes and changes in labeling that they observe.

Based on the Reviewers’ suggestions we performed these experiments. Herein, we took advantage of Y304A/G306E ETFDH mutant that impairs electron transfer from ETF and cannot substitute for the wild type (WT) gene function in ETFDH-deficient myoblasts [8]. We expressed WT and Y304A/G306E ETFDH mutant in ETFDH KO HCT116 colorectal cancer cells and confirmed that they are expressed to a comparable level (Supplementary Figure 6C). Re-expression of WT decreased proliferation, while suppressing mTORC1 signaling and increasing 4E-BP1 levels relative to control (vector infected) ETFDH KO EV HCT116 cells (Supplementary Figure 6D). In contrast, proliferation rates, mTORC1 signaling and 4E-BP1 levels remained largely unchanged upon Y304A/G306E ETFDH mutant expression in ETFDH KO HCT116 cells (Supplementary Figure 6D). Similarly, re-expression of WT ETFDH disrupted the bioenergetic phenotype associated with ETFDH loss, in contrast to re-expression of Y304A/G306E ETFDH mutant, which exhibited similar bioenergetic profiles as ETFDH KO control (Supplementary Figure 6E-F). Collectively these findings argue that the ETFDH activity is required for its tumor suppressive effects.

If nucleotide pool and labeling data are available, or can be obtained readily, this would significantly strengthen the tracing data already obtained.

We followed Reviewer’s suggestion and measured nucleotide levels. This revealed that loss of ETFDH results in increase in steady-state nucleotide pools (Supplementary Figure 2K), consistent with increased aspartate labelling and accelerated tumor growth.

References

(1) Morita, M. et al. mTORC1 controls mitochondrial activity and biogenesis through 4EBP-dependent translational regulation. Cell Metab 18, 698-711 (2013). https://doi.org/10.1016/j.cmet.2013.10.001

(2) Valenstein, M. L. et al. Structural basis for the dynamic regulation of mTORC1 by amino acids. Nature 646, 493-500 (2025). https://doi.org/10.1038/s41586-025-09428-7

(3) Appuhamy, J. A., Knoebel, N. A., Nayananjalie, W. A., Escobar, J., & Hanigan, M. D. Isoleucine and leucine independently regulate mTOR signaling and protein synthesis in MAC-T cells and bovine mammary tissue slices. J Nutr 142, 484-491 (2012). https://doi.org/10.3945/jn.111.152595

(4) Herningtyas, E. H. et al. Branched-chain amino acids and arginine suppress MaFbx/atrogin-1 mRNA expression via mTOR pathway in C2C12 cell line. Biochim Biophys Acta 1780, 1115-1120 (2008). https://doi.org/10.1016/j.bbagen.2008.06.004

(5) Jewell, J. L. et al. Metabolism. Differential regulation of mTORC1 by leucine and glutamine. Science 347, 194-198 (2015). https://doi.org/10.1126/science.1259472

(6) Tan, H. W. S., Sim, A. Y. L. & Long, Y. C. Glutamine metabolism regulates autophagy-dependent mTORC1 reactivation during amino acid starvation. Nat Commun 8, 338 (2017). https://doi.org/10.1038/s41467-017-00369-y

(7) Chen, R. et al. The general amino acid control pathway regulates mTOR and autophagy during serum/glutamine starvation. J Cell Biol 206, 173-182 (2014).https://doi.org/10.1083/jcb.201403009

(8) Herrero Martin, J. C. et al. An ETFDH-driven metabolon supports OXPHOS efficiency in skeletal muscle by regulating coenzyme Q homeostasis. Nat Metab 6, 209-225 (2024). https://doi.org/10.1038/s42255-023-00956-y